# Assessment of fNIRS Signal Processing Pipelines: Towards Clinical Applications

**Augusto Bonilauri** [1,*] **, Francesca Sangiuliano Intra** [2] **, Giuseppe Baselli** [1] **and Francesca Baglio** [2]

1. Department of Electronics, Information and Bioengineering, Politecnico di Milano, 20133 Milan, Italy; giuseppe.baselli@polimi.it
2. IRCCS Fondazione Don Carlo Gnocchi ONLUS, CADITER, 20148 Milan, Italy; fsangiulianointra@dongnocchi.it (F.S.I.); fbaglio@dongnocchi.it (F.B.)
* Correspondence: augusto.bonilauri@polimi.it

**Featured Application: In neurodegenerative diseases and neurorehabilitation, follow-up requires instrumental evidence besides clinical cognitive and motor scores. fMRI is frequently not suitable, either because patients are not eligible for an MRI or because it is prone to motion artifacts. The fNIRS technique attenuates these limitations since brain activations can be measured in a more versatile experimental setting, even if restricted to cortical activity. Therefore, the roadmap towards full clinical acceptance of fNIRS aims to provide an additional and more flexible solution to fMRI when not available or feasible, but it needs standard signal processing and protocols. This study provides comparisons of alternative processing methods in the above applicative perspective.**

**Abstract:** Functional Near-Infrared Spectroscopy (fNIRS) captures activations and inhibitions of cortical areas and implements a viable approach to neuromonitoring in clinical research. Compared to more advanced methods, continuous wave fNIRS (CW-fNIRS) is currently used in clinics for its simplicity in mapping the whole sub-cranial cortex. Conversely, it often lacks hardware reduction of confounding factors, stressing the importance of a correct signal processing. The proposed pipeline includes movement artifact reduction (MAR), bandpass filtering (BPF), and principal component analysis (PCA). Eight MAR algorithms were compared among 23 young adult volunteers under motor-grasping task. Single-subject examples are shown followed by the percentage in energy reduction (ERD%) statistics by single steps and cumulative values. The block average of the hemodynamic response function was compared with generalized linear model fitting. Maps of significant activation/inhibition were illustrated. The mean ERD% of pre-processed signals concerning the initial raw signal energy reached 4%. A tested multichannel MAR variant showed overcorrection on 4-fold more expansive windows. All of the MAR algorithms found similar activations in the contralateral motor area. In conclusion, single channel MAR algorithms are suggested followed by BPF and PCA. The importance of whole cortex mapping for fNIRS integration in clinical applications was also confirmed by our results.

**Keywords:** continuous wave functional near-infrared spectroscopy; rehabilitation monitoring; brain activation mapping; motor tasks; functional near-infrared signal processing; movement artifact removal; hemispheric hemodynamic response; clinical fNIRS translation

## 1. Introduction

For more than two decades, functional Near-Infrared Spectroscopy (fNIRS) has become a valuable tool for non-invasively estimating the Hemodynamic Response Function (HRF) and mapping cerebral cortex activations [1]. The phenomenon of HRF had been observed many decades earlier in invasive experiments investigating the fine functional mapping of the exposed cortex. However, in that context, the HRF was just one of the

many intrinsic or dye-enhanced optical responses available to investigators [2–5]. The feature which allows HRF to be accessible to fNIRS and other non-invasive transcranial measurements is the critical variation of perfusion to even modest increments in the local oxygen consumption. This is probably needed to compensate for transport delays from vessels to cells and is driven by powerful neuro-vascular coupling mechanisms [5]. Briefly, neuro-vascular coupling actively enhances the regional cerebral blood flow and volume. A peak in oxygenated hemoglobin concentration $[HbO_2]$ having a time delay of 4–5 s after neuronal activation is shown. Venous outflow increases, causing a drop in deoxygenated (alias, reduced) hemoglobin concentration $[HbR]$. Next, the hemodynamic equilibrium recovers its baseline within 10–12 s. Accordingly, HRF-related signals are powerful, yet indirect markers of neural activity with a fair spatial resolution (order of 10 mm) but limited time resolution (order of seconds, compared to the 1 ms of neural action potentials). Nonetheless, functional imaging has gained a prominent position in studying brain anatomic-functional organization, functional and practical connectivity, and their damage associated with neural pathologies [6].

The first alternative approach to fNIRS for monitoring regional blood volume changes was based on blood labeling radiotracers in Single Photon Emission Computed Tomography and Positron Emission Tomography [7]. Despite the limitation in both space and time resolution of nuclear imaging, these pioneering studies paved the way to the actual application of functional imaging. The advent of functional MRI (fMRI) [8] offered better space and time resolution, as well as complete non-invasiveness and improved signal-to-noise (SNR) ratio. Indeed, fMRI is currently the gold standard in the non-invasive investigation of neural activity, even if the unique quantification capabilities of nuclear medicine should not be forgotten. fMRI is based on the blood-oxygen-level-dependent (BOLD) contrast, which reflects up to 4–5% of the overall anatomical contrast. The BOLD signal is due to the washout, which augments the local magnetic field homogeneity and, ultimately, the signal level due to its paramagnetic properties. This method is characterized by a good spatial resolution (i.e., order of 10 mm) in functional mapping without any depth resolution limits due to the ability to map the whole cortex as well as subcortical and spinal gray matter layers. However, particularly in clinical setups, fMRI presents applicative short comes of MRI scanner complexity and costs, difficulties with non-cooperating subjects, claustrophobia, and high sensitivity to motion artifacts. In addition, the high magnetic field environment causes problematic multi-modal integration with EEG.

Conversely, fNIRS presents clinical application potentials which may overcome some of the fMRI limitations. With respect to poorly cooperating subjects, fNIRS cannot be performed, which represents a real opportunity to overcome the motion artifact sensitivity and practical or environmental eligibility, such as the presence of mental implants or suffering from claustrophobia [9,10]. This technology measures attenuation changes of at least two NIR-wavelengths around the isosbestic point at 805 nm, where $HbO_2$ and $HbR$ absorbances are equal. In this way, depending on the employed fNIRS technology, either absolute or relative concentration changes can be measured and employed to estimate HRF associated with functional activation. The main advantage of using fNIRS in clinical applications compared to fMRI is its higher tolerability. Indeed, functional measurements are acquired in an open environment and by only employing an EEG cap coupled to NIR-optodes, namely, either sources or detectors. This approach allows us to map brain activity according to measurement channels given by neighboring source and detector pairs, hence having a spatial resolution of 3–4 cm. Although movement related artifacts are still an issue, they are reduced to the sole coupling between a subject's scalp and fNIRS-cap, thus avoiding the rigid head binding imposed by a fMRI setup. As a result, fNIRS functional and resting-state acquisitions can be also performed in impaired subjects who cannot prevent head movements. Patients' watching and care is also not restricted as in an MRI tunnel, and claustrophobic distress is avoided.

Conversely, the main fNIRS limitation is related to its surface approach opposed to the volumetric sensitivity of fMRI. Hence, only the sub-cranial cortex is sensed by NIR-light,

leaving out activations in the mesial cortex, in deep scissures, and in all the deep gray matter nuclei. Interestingly, the cortex areas sensed by fNIRS and EEG are almost the same, and the two systems are often integrated with negligible interference. This potential, though beyond the scope of this study, further increases the interest in clinical fNIRS applications [11].

Despite the above technical and portability advantages, fNIRS translation to clinics is still hindered by its limited spatial resolution and SNR due to motion artifacts (MAs) and intrinsic physiological interference over the acquired signal. Moreover, with reference to the signal preprocessing and analysis, a standard procedure is still missing due to a wide heterogeneity of available instrumentation and research objectives, which translates into a poor impact on interpretation and reproducibility of results [12]. Consequently, fNIRS is still not approved for stand-alone diagnostic and therapeutic procedures. Therefore, careful preprocessing of the signal is required prior to HRF estimation. As mentioned before, this process is still opened to active biomedical research and is far from the established standards required by clinics. As will be discussed in Section 2.2, the small contribution of $HbO_2$ and $HbR$ absorbance in the brain vessels, compared to all the other tissues (such as the scalp, skull, liquor, and meninges) makes fNIRS highly sensitive to motion-related changes and coupling of optodes. Hence, a major problem in fNIRS signal processing is associated with the detection and reduction of related MAs. In addition, physiological confounding factors (PCFs) not related to neurovascular coupling are generated by vascularized tissues (scalp and meninges, mainly), which respond to systemic cardiovascular hemodynamics and regulation, including the pulse, respiratory-related modulations, and slower vasomotor mechanism such as Mayer waves, hence possibly misinterpreting results associated with brain activation [13].

The problem with MAs and PCFs can be addressed by instrumental solutions, namely, by employing fNIRS technologies able to estimate absolute concentrations of $HbO_2$ and $HbR$, e.g., Time-Domain fNIRS, which can resolve the penetration depth according to the time-of-flight (TOF) of photons reaching the cortex [14], or Frequency-Domain NIRS that employs intensity-modulated and phase-sensitive detection based on the mean TOF [15]. Other strategies provide adaptive cancellation of artifacts based on ancillary sensors to regress superficial confounding factors from the fNIRS signal. Among these approaches, and considering the clinical experimental setting, the best balance could be represented by the short-separation channel regression [16,17], whose performances can be additionally enhanced by movement sensors and hemodynamic sensors [18–20].

However, most commercial fNIRS instruments are based on Continuous Wave devices (CW-fNIRS) and are often not coupled to a separate artifact sensing system. Therefore, the present study addresses this condition, which leaves all the artifact removal burden (and uncertainties) to preprocess the raw fNIRS signals. In this perspective, some of the significant artifact reduction strategies will be presented, and results relevant to several combinations of them will be compared in a group of healthy volunteers performing a hand grasping motor task. Among the prospective benefits of this study, the presented results can be taken as a justification for adopting specific preprocessing and analysis pipelines because of further clinical studies, which may not employ a separate artifact sensing system.

It is worth remarking that a thorough review of published fNIRS cleaning algorithms is out of the study scope. In addition, experimenting with all possible combinations of options in the various preprocessing steps might lead to many comparisons. Conversely, we used only a small set of the main MA removal algorithms [21–25], adding some variants explained in the Methods section, with a total of eight algorithms. The subsequent bandpass filtering and principal component analysis (PCA) steps were uniformly repeated in the eight compared pipelines. Despite the necessarily limited examples, we provide quantitative comparisons relevant to the energy reduction of each step in each pipeline and the final HRF outcome, aiming at giving the reader both a theoretical and an applicative perspective.

## 2. Materials and Methods

In this section, we present the processing steps required for computing $HbO_2$ and $HbR$ concentration changes in CW-fNIRS data, motivating them by discussing the components, artifacts, and confounding factors of the raw optical density (*OD*) signal. A concise description of the compared algorithms is then given. Finally, the experimental setup is described, along with the comparison criteria.

### 2.1. Conversion to Optical Density and Concentration

According to the Beer–Lambert Law, absorbance is evaluated by the optical density (*OD*):

$$OD(\lambda, t) = \log\left(\frac{I_0(\lambda)}{I(\lambda, t)}\right) = \log(I_0(\lambda)) - \log(I(\lambda, t)) \tag{1}$$

where $I_0(\lambda)$ is the non-attenuated light intensity and $I(\lambda, t)$ is the sensed intensity at time $t$ after passing through a biological sample. In the context of NIRS, this equation needs to be extended to the Modified Beer–Lambert Law (MBLL), introducing a scattering dependent parameter to take into account the optical properties of diffusive media such as biological tissues and the increase in photon pathlength compared to source–detector separation [26] (see also Section 2.2). Namely, most commercial CW-fNIRS systems employ two NIR-wavelengths placed within the optical window on opposite sides of the isosbestic point (805 nm) for maximizing $HbO_2$ vs. $HbR$ discrimination [27]. Hence, the MBLL equation considering only $HbO_2$ and $HbR$ chromophores is represented by the following equation

$$\Delta OD_{i,j}(\lambda, t) = L_{i,j} \cdot DPF(\lambda) \cdot (\varepsilon_{HbO}(\lambda) \cdot \Delta[HbO_2] + \varepsilon_{HbR}(\lambda) \cdot \Delta[HbR]) \tag{2}$$

where $L_{i,j}$ is the $i$th-source to $j$th-detector length measured at the scalp surface, and $\varepsilon_{HbO}(\lambda)$ and $\varepsilon_{HbR}(\lambda)$ are the respective oxy- and deoxy-hemoglobin molar extinction coefficients.

$DPF(\lambda)$ is the differential pathlength factor, a unitless scalar parameter whose $L_{i,j}DPF(\lambda)$ product accounts for the longer travel of scattered photons. In CW-fNIRS the $DPF(\lambda)$ cannot be estimated directly; hence, tabulated DPFs derived from other studies from TD-fNIRS, FD-fNIRS, ex-vivo and in-silico simulations are considered [28,29]. As a result, CW-fNIRS studies cannot be said to be quantitative. Nonetheless, the appropriate use of these factor permits an even scaling in the presence of different $L_{ij}$ distances and of subjective differences in $DPF(\lambda)$, which is strongly age dependent [28,30]. An absolute quantification of HRF is beyond the scope of CW-fNIRS, which conversely addresses the statistical significance of HRF deflections above confounding factors and noise. In addition, in CW-fNIRS and limited to the target brain vessels, the scattering dependent parameter in the traditional formulation of MBLL can be neglected since optical quantities are computed differentially and scattering is limited compared to absorption effects due to only $HbO_2$ and $HbR$ (Equation (2)).

Finally, the MBLL for $HbO_2$ and $HbR$ results in a system of two linear equations in two unknowns, namely, $\Delta[HbO_2]$ and $\Delta[HbR]$ concentration changes, with solution

$$\begin{bmatrix} \Delta[HbR] \\ \Delta[HbO_2] \end{bmatrix} = \begin{bmatrix} \varepsilon_{HbR}(\lambda_1) \; \varepsilon_{HbO_2}(\lambda_1) \\ \varepsilon_{HbR}(\lambda_2) \; \varepsilon_{HbO_2}(\lambda_2) \end{bmatrix}^{-1} \begin{bmatrix} \frac{\Delta OD(\lambda_1)}{DPF(\lambda_1) \cdot L} \\ \frac{\Delta OD(\lambda_2)}{DPF(\lambda_2) \cdot L} \end{bmatrix} \tag{3}$$

In this work, we selected the *DPF* value according on the work of Scholkmann and Wolf [28], who proposed a general equation for the computation of age- and wavelength-specific values based on published data. We are aware that the proposed equation was designed for frontal cortex optical data. However, even if the employed fNIRS probe addresses the whole superficial cortex (details are provided in Section 2.8), it is generally assumed that this aspect does not drastically affect pre-processing and analysis since relative (not absolute) functional activation is addressed.

## 2.2. Comments on Artifact Sources

So far, the MBLL was considered as if the absorbance of tissues other than brain vessels (BVs) was constant and was accordingly omitted in its differential expression (Equation (2)). Conversely, motion dynamics $m(t)$ cause large changes in the distance $\Delta \overline{L}_{tissue}$ travelled through a given tissue; hence, measured $\Delta OD(\lambda, t)$ quantities are strongly affected by underlying tissue distribution. In addition, the traditional fNIRS HRF is highly contaminated by both extracerebral and cerebral components not directly related to evoked brain activity and related to systemic hemodynamics $s(t)$ [13]. Therefore, these changes in the effective distance of photon travel are multiplied by the relevant linear attenuation $\mu_{tissue}(\lambda) = \varepsilon_{tissue}(\lambda) \cdot c_{tissue}$, which is here approximately assumed as a constant specific to the tissue concentration, with the only exception of the blood compartment.

$$\Delta OD(\lambda, t) = DPF(\lambda) \cdot \Delta L_{tissue} \cdot \mu_{tissue}(\lambda) + n(t) \tag{4}$$

which in turn is subdivided according to tissues encountered along the photon path

$$\Delta OD(\lambda, t) = DPF(\lambda) \cdot \left\{ \begin{array}{c} \Delta L_{BV} \cdot \mu_{BV}(\lambda) + \Delta \overline{L}_{skin}(m(t)) \cdot \mu_{skin}(\lambda) \\ + \Delta \overline{L}_{bone}(m(t)) \cdot \mu_{bone}(\lambda) \\ + \Delta \overline{L}_{liquor}(m(t)) \cdot \mu_{liquor}(\lambda) \\ + \Delta \overline{L}_{cells}(m(t)) \cdot \mu_{cells}(\lambda) \\ + \Delta \overline{L}_{blood}(m(t), \ s(t)) \cdot \mu_{blood}(\lambda) \end{array} \right\} + n(t) \tag{5}$$

The high sensitivity of effective distances $\Delta \overline{L}_{tissue}$ to motion $m(t)$ relates to geometrical variations in source–detector coupling, even with minimal changes in the relative source–detector distance and orientation. Namely, since absorption variations due to brain perfusion changes are mainly referred to the *BVs* within the cortical layer, quantification of relative changes in $HbO_2$ and $HbR$ concentrations can be corrected by introducing a Partial Volume Factor $PVF(\lambda)$, considering the Partial Pathlength Factor $PPF(\lambda) = DPF(\lambda)PVF(\lambda)$ to account for variations of optical pathlength over only the cortical layer [30,31].

Classically, the source–detector coupling volume is depicted as a photon path whose ends are the source and the detector, while the diffused photons enter the tissues, touching the cortex in its deepest midway region. This schematically, yet clearly, shows the huge weight of tissues other than BVs. Consequently, large unpredictable changes in the presence of head acceleration forces and contraction of cranio–facial muscles can be expected. Furthermore, several elements concur in the amplification of the unwanted artifactual changes vs. the BVs pathways: (i) the effective lengths of surface tissues are heavily weighted by the higher illumination density in external layers compared to the cortex; (ii) among them, bone displays the highest linear attenuation, which is not sufficient to hinder transcranial fNIRS (thanks to the NIR optical window), yet is several times higher than hemoglobin [32].

As for PCFs, according to Equation (5), the absorbance of blood compartments other than BVs is lumped into a single term, which should mainly include skin and meninge vessels responding to systemic circulation drives $s(t)$. In addition, the extent of the $s(t)$ effects on BVs themselves (not shown in Equation (5)) is still an open question. Irrespective of the driving mechanisms, the general criterion to separate PCFs is the hypothesis of a generalized action on all optodes. Hence, all cleaning strategies share the removal of common-mode components (details are provided in Sections 2.5–2.7).

## 2.3. The Proposed Order of CW-fNIRS Processing Steps

The unprocessed $\Delta OD$ signals at the two wavelengths were separately processed prior to the application of MBLL. This choice was motivated by the unpredictable effect of biases on both $\Delta OD$ signals while solving the MBLL (Equation (3)), which conversely represents a mandatory linear operation of each single processing pipeline.

After a quality check of the raw signals (described at Section 2.4), the first actual signal processing step performs MA reduction since it considers non-linear and time-variant

operations over the $\Delta OD$ time course. In fact, all typologies of MA algorithms share the detection of outlying features localized in time and their correction. Accordingly, our suggestion is that no other processing algorithm should be applied prior to this step since any previous cleaning or filtering might hinder the detection phase. Hence, the second step was linear bandpass filtering (BPF), next followed by common-mode reduction of physiological interference by principal component analysis (PCA). In principle, the linearity of these steps should permit their full interchangeability. However, we preferred to filter data before PCA since the latter method works on the global components of the signal (see Section 2.7).

Finally, after MBLL application, the HRF can be estimated either by block averaging of $\Delta[HbO_2]$ and $\Delta[HbR]$ across trials or through linear estimation models such as the General Linear Model (GLM) [29]. While both approaches allow us to infer cortical activation and test its statistical significance, block averaging does not impose any temporal constraints over the shape of HRF. Conversely, the GLM approach allows us to model the measured response as a linear combination of effects and confounding factors. In the results section, we provide graphical and numerical evaluation of activation according to both approaches.

Pre-processing and analysis were conducted in Matlab R2018b (The MathWorks, Inc., Natick, MA, USA) through user-defined scripts integrating available functions of Homer2 developer's version [33] and NIRS Brain AnalyzIR Toolbox [34]. In conclusion, the proposed pipelines follow these consecutive stages: (1) MA reduction (herein eight variants); (2) bandpass filtering; (3) reduction of physiological interference by PCA; (4) MBLL for $\Delta[HbO]$ and $\Delta[HbR]$ computation; (5) block averaging and significance tests of single channel responses.

### 2.4. Channel Exclusion Criterion

Ahead of the signal processing pipeline, measurement channels presenting a low SNR must be identified and excluded from further analysis, and participants presenting an overall low SNR across measurement channels must be excluded from further analysis in order not to lead to misinterpretation of group-level results. In line with other studies [12,35], we considered SNR to measure the local coefficient of variation above a given threshold $Th$:

$$CV\%_{tw}(t) = \frac{STD_{tw}(t)}{AVG_{tw}(t)} \cdot 100\% > Th \tag{6}$$

where subscript $tw$ indicates the duration of the time window $[t - tw/2; t + tw/2]$ in which the standard deviation (STD) and average (AVG) statistics were computed. Namely, this time window is set to all time periods where a functional task is performed (i.e., overall duration of experimental blocks), hence obtaining the overall SNR of measurement channels. Threshold $Th$ was set to 7.5% within the range of previously published works [12,35]. Conversely, the former step of excluding participants from analysis due to an overall low SNR can be performed according to visual inspection of the dataset. Besides visual inspection, we decided to also perform this step automatically by labeling low SNR datasets when $CV\%_{tw}(t) > Th$ exceeded 10% of the overall number of measurement channels across wavelengths. In this work, five out of 23 participants were excluded due to low channel-wise SNR across the overall measurement channels (further details regarding the dataset are presented in Section 2.8).

### 2.5. MA Reduction Algorithms

In general, MA reduction algorithms involve an identification step to test the presence of outliers and label MA windows, which will be corrected in a second correction step. Hence, MA reduction algorithms are time-variant (most often in on–off fashion) and non-linear due to thresholding methods employed in the identification step.

Moreover, most algorithms separately label MAs in single channels (SCs). However, the possible correlation of MA occurrence among channels is an open question. Hence, we also tested multi-channel (MC) variants in which the SC labelling was propagated to all

channels according to "OR" logical operator. In brief, MA labeling was extended to all time windows where at least one of the signals showed outlier features. Our interest in such variants was two-fold: (i) a large increase in labelled tracts passing from SC to MC would indicate that MA timing is sparse through channels, thus contrasting the hypothesis of common MA sources; (ii) the level of disruption in the useful HRF, which might be caused by an overcorrecting MC strategy.

The correction algorithms of MA labelled windows were based on the reduction of outlier features by substitution of the raw signal with common-mode dynamics found by targeted PCA (tPCA, see Appendix A.3) [23] and spline interpolation [21]. Starting from the original version of the Movement Artifact Reduction Algorithm (MARA) strategy proposed by Scholkmann et al. [21], we also tested the hybrid version proposed by Jahani et al. [22] to simultaneously detect baseline shifts and spike artifacts. As illustrated in the Appendix A, the MA detection and the correction algorithms are found in the current literature. However, we tested several combinations of identification and correction methods, some of which not considered by the original works, and according to both the SC and the MC labeling. We also considered two additional algorithms, namely, the Temporal Derivative Distribution Repair (TDDR) [24] and a wavelet-based method [25], which conversely apply signal decomposition methods to the overall signal trend rather than acting only on specific MA windows. In conclusion, these MA algorithms paired to SC- vs. MC-labelling and correction methods led to eight different variants, successively considered pipelines, to discuss the results of HRF estimation. These algorithms are labelled as follows: SC-MARA-Spline, SC-MARA-tPCA, MC-MARA-Spline, MC-MARA-tPCA, HybridMARA-Spline SG, HybridMARA-tPCA SG, TDDR, and Wavelet.

### 2.6. Bandpass Filtering

Despite the large MA correction of any of the above algorithms, linear bandpass filtering is still a good practice for reducing very slow drifts and high frequency noise. In this study, the standard choice of 0.01–0.2 Hz bandwidth was adopted [36]. Although this step might be trivial, we were interested in reporting how the amount of power reduction introduced by filtering depended on the previous MA reduction step.

Moreover, both the high-pass $f_{HP}$ and the low-pass $f_{LP}$ frequency cutoffs deserve some critical evaluation relevant to the experimental design (details in Section 2.8) and to the final statistical assessment of HRF, respectively. In block-design experiments with a repetition frequency $f_{BD} = (T_{task} + T_{rest})^{-1}$, having $f_{BD} > f_{HP}$ is a requirement to not filter out the addressed HRF task-driven oscillations. The motor task adopted for this study considers $T_{task} = 10$ s and $T_{rest} = 20$ s, leading to $f_{BD} = 0.033$ Hz $\gg f_{HP}$. In the case of a lower margin, it might be necessary to high-pass filter the HRF regressors, prior to statistical analysis by the generalized linear model (GLM), or inverse-filter the HRF shape extracted by block-averaging, to recover the unfiltered HRF shape.

Low-pass filtering at $f_{LP} = 0.2$ Hz improves SNR. Apparently, the elimination of oscillations shorter than 5 s should have no drawbacks given the slower HRF dynamics. This holds if the final statistics are based on the block average. Conversely, detrimental effects should be considered in case of GLM analysis, where statistical significance relies on the whiteness of residuals and whose requirement is not satisfied by the low-pass filter smoothing [37,38].

### 2.7. Reduction of Physiological Interference by PCA

PCA and correction by the removal of the largest components is one of the earliest fNIRS signal conditioning methods [39]. Several studies have compared PCA to other MA reduction algorithms; however, its limits in treating such unsmooth and heterogeneous artifacts have been pinpointed (see the discussion on tPCA in Appendix A.3). Conversely, PCA is fully valid in addressing the reduction of common-mode PCFs due to hemodynamic activities other than the HRF [40]. The working principle consists of enhancing the contrast of localized task-related activations by removal of the largest components of

PCA, which are considered global effects common to all measurement channels. Indeed, considering the more extended framework presented in Section 2.2, fNIRS measurements are highly affected by extracerebral and non-evoked cerebral activity, making only the BPF operation insufficient to estimate the corresponding HRF [13], especially for Mayer waves and respiratory oscillation due to partial overlap with the frequency content of HRF [41].

Alternative methods, not applied in this study, that do not involve auxiliary measurements of extracerebral and systemic activity are based on the extraction of a common-mode component as a median of all signals (or all signals in one hemisphere), next regressing it out from each channel (or each channel of the opposite hemisphere) [12]. Additionally, the impact of slow-varying physiological oscillations and scalp-related superficial confounds can be reduced by employing techniques such as Discrete Cosine Transform, pre-coloring, and pre-whitening methods directly in subject-level analysis [42,43]. The critical parameter in PCA cleaning is the amount of power to be deleted, which in the present study was set to remove PCA components by up to 80% of the variance in the data, similar to other studies [44]. The PCA method is particularly suited to remove physiological interference over the considered experimental dataset, since we considered an fNIRS probe that almost covers the whole subject scalp surface (details are presented in Section 2.8). Hence, the largest PCA components do identify global variations of physiological oscillations, leaving only 20% in the variance of the data as localized HRF-related activity.

### 2.8. Subjects and Experimental Set-Up

The presented pilot analyses were carried out on a group of 23 healthy young adults (age 28.3 ± 4.0 years) aiming at setting up a processing pipeline suited to clinical applications employing CW-fNIRS instrumentation. Experimental data were acquired at IRCCS Fondazione Don Carlo Gnocchi, Milan. The instrument was the NIRScoutX 32 × 32 (NIRx Medizintechnik, Berlin, Germany). This system performs functional measurements at 760 and 850 nm wavelengths and employs LEDs as the source technology and avalanche photodiodes as detectors. The head cap consisted of 32 detectors and 32 sources locations, making a total of 102 measurement channels between them with a resulting sampling frequency of 1.9531 Hz. The mean source–detector distance, referring to the virtual co-registration of this configuration in the Colin27 atlas in AtlasViewer software package [45], was 3.46 cm (Figure 1).

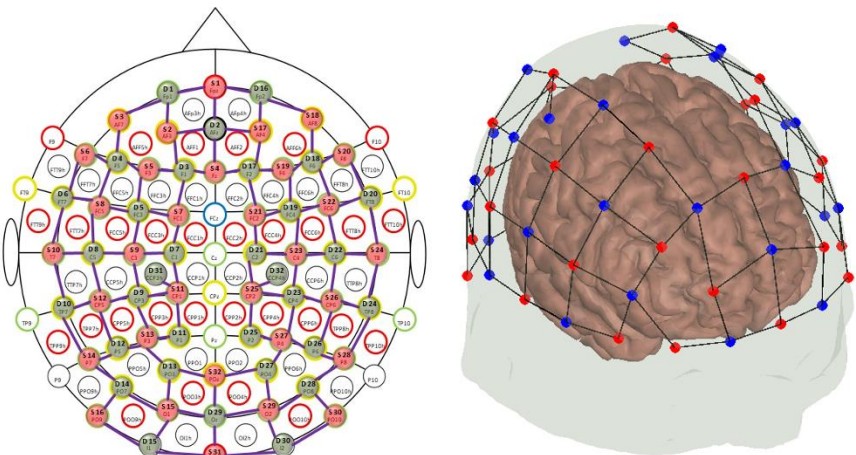

**Figure 1.** Employed fNIRS probe configuration in the current study. (**Left**) Red and green dots indicate the respective location of LED sources and fiber optics detectors according to the 10–10 reference system labeling. (**Right**) Virtual co-registration of source (red dots) and detector (blue dots) locations over the Colin27 anatomical template in the AtlasViewer software package [45].

The functional task consisted of a motor-grasping task where participants were asked to alternatively move their left or right hand by repeatedly squeezing sponge balls placed

in their palms. Together with the similar version of finger-tapping, this functional task is widely used in fNIRS literature for assessing cortical activation associated with motor functions over motor and somato-sensory areas mapping the hand [46]. A block-design paradigm was adopted, and a schematic representation of the timing is presented in Figure 2. Participants had to sit still in a dim light room watching a stimulus presentation screen with a cross placed at its center, which represented the resting condition. Conversely, the task condition was the repeated left vs. right grasp and release of the sponge ball when the cross blinked at 0.8 Hz over the left vs. right side of the screen. The task condition lasted 10 s followed by 20 s of the resting condition, while stimulus presentation was randomized between the hand movement and repeated 10 times. No adverse conditions were simulated during the data acquisition.

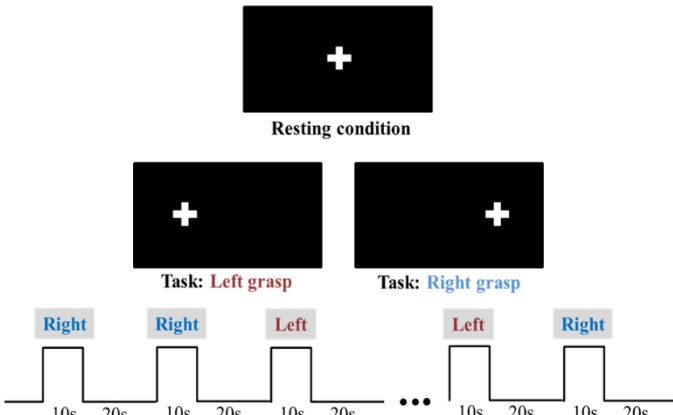

**Figure 2.** Schematic representation of the block-design motor grasping task (**bottom**) and indicative illustration of blinking cross over the stimulus presentation screen according to task vs. resting conditions (**top**).

### 2.9. Signal Correction Metrics

The proposed comparison of pipeline variants (changing the MA reduction with the same filtering and PCA) suffers from a lack of ground-truthing. A possible way to systematically compare the results of different pipelines includes an assessment with Receiving Operating Characteristics (ROC) curves of simulated data [34]. This process is highly recommended to simulate HRF activation over resting-state data, which were not available during data acquisition. Therefore, given this limitation, we monitored the effects in energy reduction exerted by each step and its variants.

The Energy Relative Decrease (ERD) was tabulated over optical density at each step of the processing pipeline—namely, at MA rejection, BPF, and PCA steps—with respect to the starting energy of the un-processed optical density data as a percentage of energy reduction. This concept can be summarized the following equation

$$ERD_{step}\% = \frac{E_{f,step} - E_i}{E_i} 100\% \tag{7}$$

where $E_{f,step}$ and $E_i$ indicate the respective mean total energy of the $\Delta OD$ signals ahead of the single pre-processing step and prior to MA reduction.

The performances of the BPF filter step in the 0.01–0.2 Hz frequency range were additionally tabulated according to the power ratio between filtered and unfiltered signals. More precisely, the Power Ratio within-band ($PR_{wb}$) and outside-band ($PR_{wb}$) (i.e., passband and stopband, respectively) was computed as a percentage ratio between the mean Power Spectral Density (PSD) ahead and prior to the BPF step. This concept can be summarized in the following equation

$$PR_{wb}[\%] = \frac{F_{PSDwb}}{U_{PSDwb}} 100 \tag{8}$$

$$PR_{ob}[\%] = \frac{F_{PSDob}}{U_{PSDob}} 100 \tag{9}$$

where $F_{PSD}$ and $U_{PSD}$ indicate the respective filtered and unfiltered PSD of $\Delta OD$ signals.

Finally, the resulting significant activations by block-average and group-level analysis were compared, which provided an overall quality assessment since the expected activations presented high confidence in the simple motor task on healthy subjects. Hence, the block-average outcomes of the differently cleaned fNIRS signal were plotted and visually compared. The block average step was performed with Homer2 software (*hmrBlockAvg* function), while group-level maps of activation were computed by means of the NIRS Brain AnalyzIR Toolbox as a Mixed Effect Model (task condition as a Fixed effect, while intercept was a random effect using subjects as a grouping variable). It is worth noting that for performing group-level analysis and hence subject-level GLM analysis, the filtering step was modified to HPF instead of BPF according to the motivations explained in Section 2.6.

A synthetic measure of SNR was extracted from block-averaged responses across subjects according to the following equation

$$SNR\,[dB] = 10 \log_{10}\left(\frac{E_{BA}(t_{activation})}{E_{BA}(t_{baseline})}\right) \tag{10}$$

where $E_{BA}(t_{activation})$ and $E_{BA}(t_{baseline})$ indicate the respective total energy of the block-averaged response over equal time periods where we expect the maximum of activation vs. return to baseline concentrations. Namely, $t_{activation}$ was set in the 6–12 s range after the stimulus onset, with $t_{baseline}$ in the 24–30 s range.

## 3. Results

The results of fNIRS signal processing are given relevant to the group of 23 healthy young adults (HYA). Examples of signals before and after each processing step, comparing the diverse MA removals, are plotted. The effect of the successive steps is statistically presented, focusing on the progressive reduction of signal energy through the cleaning process. Finally, single subject examples and group average results are given relevant to the HRF output by the various processing pipelines.

### 3.1. Motion Artifact Results

Limited to Hybrid-MARA and SC-MARA and MC-MARA, the detection of the MA windows is compared in Figure 3. The percentage of the signal labelled MA was similarly low in both Hybrid-MARA (1.10–0.90%, median–IQR), which is a SC-method, and SC-MARA (0.32–0.42%, median–IQR). The two methods, conversely, diverged concerning the average length of labeled windows (1.73–0.57 s for HybridMARA, 3.15–0.36 s for SC-MARA) and the former showed a more focused MA detection, due to the combination of both the local std (equal to MARA) and a gradient criterion.

With respect to MC-MARA, it is worth recalling that this variant was implemented in this study to test the possible crosstalk of MAs among fNIRS channels. The percentage of the signal labelled MA, after the MC "OR" logical operator, was abruptly increased (6.46–5.85%, which was about 4-fold compared to SC), thus showing a large independence among such artifacts in the various channels. The increased length of labelled tracts (3.84–1.79 s) is a trivial consequence of this observation. This result does suggest that SC MA reduction methods are well justified.

It is also worth noting that boxplots provided in Figure 3 only refer to the identification step of the HybridMARA, MC- and SC-MARA algorithms. The results are identical between the HybridMARA-tPCA SG vs. HybridMARA-splineSG algorithms, as well as the MC/SC-MARA-tPCA vs. MC/SC-MARA-spline versions. Conversely, no boxplots were provided for the Wavelet and TDDR methods since these algorithms do act on the overall trend of the signal instead of separate segments. Therefore, computing the motion artifact length and percentage over processed signals will give results close to 100% and the overall signal

length. As a consequence, only three boxplots per motion artifact feature were presented in Figure 3.

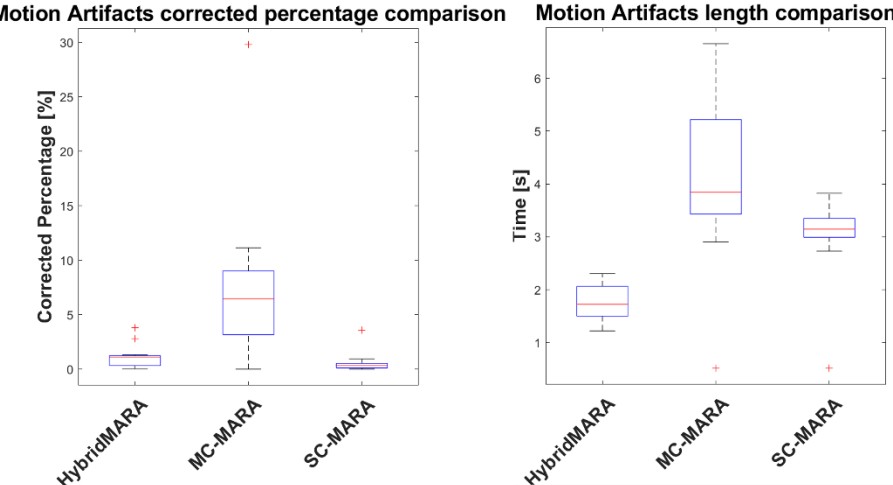

**Figure 3.** Boxplot comparison of averaged MA percentage (**left**) and temporal length (**right**) across subjects and measurement channels. Cross symbols outside interquartile ranges indicate data outliers.

In Figure 4, an example of MA reduction across the eight algorithms is shown comparing the signal before (i.e., blue line) and after MA correction (i.e., red line). For the first six methods, the labelling of MA windows (1 for MA detected, yellow) is also added. This last aspect is not presented for the last two since they analyze the overall features of the signal.

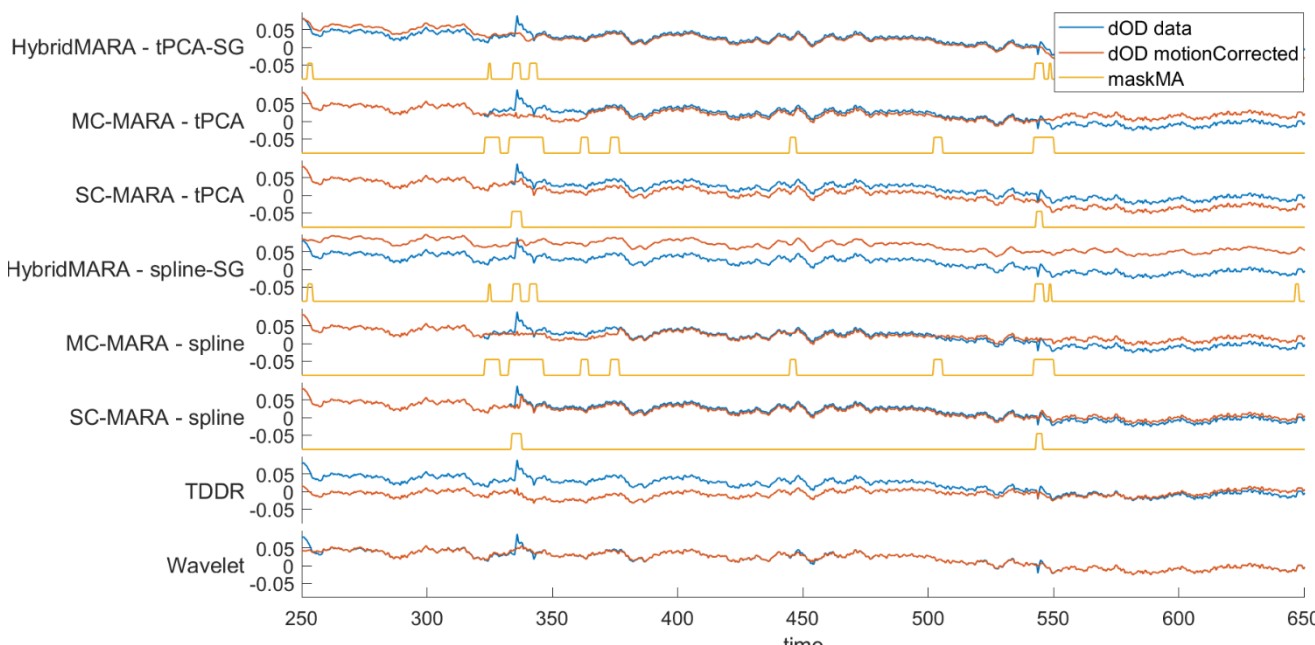

**Figure 4.** Single-channel graphical representation of the effect of MA reduction over $\Delta OD$ signals across pipelines. $\Delta OD$ signals are represented with respect to the unprocessed signal (blue line-input signal) and after MA reduction (red line-output signal). Additionally, signal tracts labelled as MAs are presented as squared waves (yellow line).

Largely different behaviors in baseline drift correction are also shown, which evidently emphasizes the need for the subsequent BPF step, with respect to the HPF action. Interestingly, all methods efficiently detected local anomalies, such as the MA spike at about 340 s. However, large differences are shown in the removal action. Namely, all six MARA

variants appeared to exert heavy interpolation, which might eliminate useful signals if occurring at high rates throughout the whole recording. Obviously, this risk might be hugely amplified in MC logic. A more conservative cleaning is conversely shown by the TDDR and the Wavelet methods.

### 3.2. Bandpass Filtering Results

The same example provided in Figure 4 but referring to the BPF step is shown in Figure 5. Clearly, the HPF does eliminate the baseline drifts not removed by the MA corrections due to their local and non-linear action. Results relevant to the LPF smoothing are similar for all eight pipelines. However, the amount of removed HF noise presents interesting differences. Namely, only the two Hybrid-MARA approaches appear to have had significant smoothing in the previous MA reduction step when applying SG smoothing as a last step of the algorithm scheme.

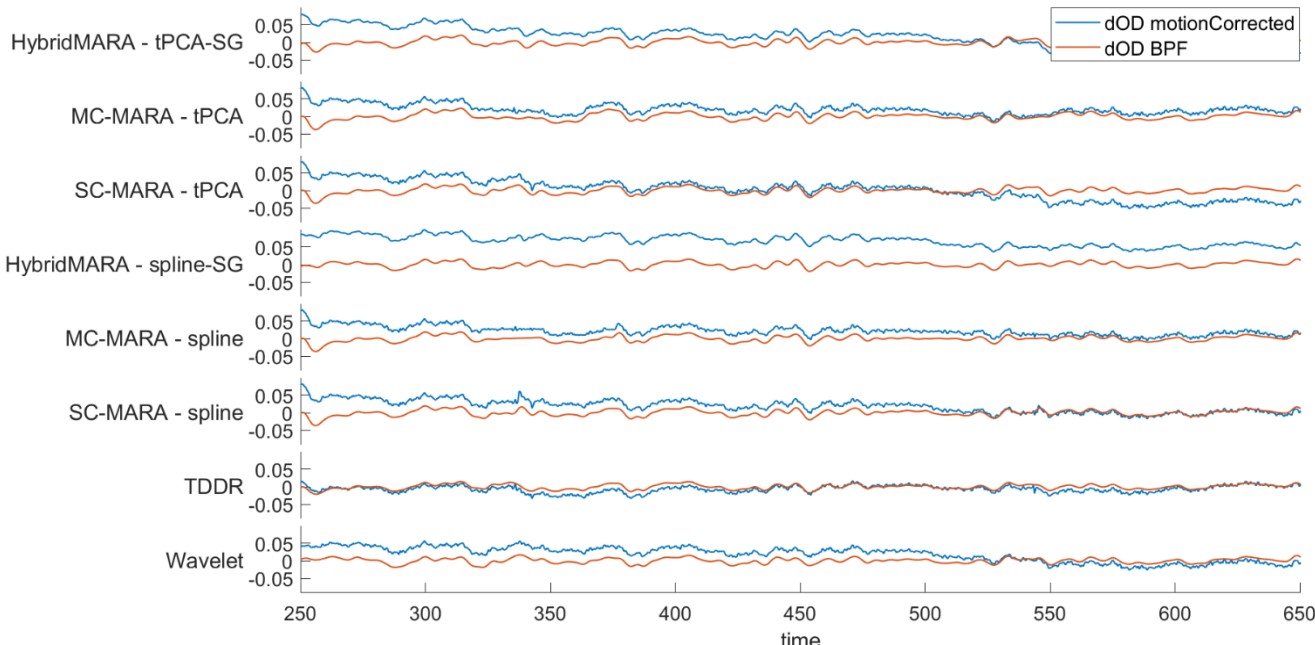

**Figure 5.** Single-channel graphical representation of the effect of BPF over $\Delta OD$ signals across pipelines. $\Delta OD$ signals are represented with respect to the MA-corrected signals (blue line-input signal) and after BPF step (red line-output signal).

It is noteworthy that the wavelet method, which reduced MAs, did not perform effective filtering in both the stop bands: LF, for baseline drift reduction, and HF, for signal smoothing. This is a direct consequence of the data-driven setting of the normality range of the method, with no a-priori definition of signal vs. noise scales. Therefore, the noisy scales of slow baseline drifts and fast HF noise were preserved. Removal of unwanted bands was left to the next BPF phase, which confirms the need of this further step.

The modification of the spectral content performed by the BPF is analyzed in Table 1, where the I/O power ratio (output over input %) is shown both in the passband (within band, $PR_{wb}$) and stopband (outside the band, $PR_{ob}$), within and outside the $f = [0.01 - 0.2]$ Hz range. Due to the BPF linearity, I/O differences should be attributed to the method-specific impact of LPF and HPF transition bands in the specific algorithm. Apparently, all methods, with the only exception of TDRR, displayed a large power content at the edges of the passed bandwidth, where power loss occurred, compared to the ideal BPF. This led to 20% or more power loss in the ideal passed bandwidth. Such non-ideal loss was reduced to about 10% in TDRR. As for $PR_{ob}$, the ideal stopband behavior should be a 0% power ratio, which should be best approximated away from the cutting edges. Therefore, the larger values above 1% displayed by the Hybrid-MARA approaches should indicate the

presence of significantly greater power in the outside band transition regions. These results suggest that the BPF order and design could be profitably adapted to the previous MA reduction step.

**Table 1.** Mean (std) Power Ratio at the passband ($PR_{wb}$) and stopband ($PR_{ob}$) in the $f = [0.01 - 0.2]$ Hz range across subjects and measurement channels. The leftmost column indicates the adopted MA reduction algorithm prior to the BPF step.

|  | $PR_{wb}$ | $PR_{ob}$ |
|---|---|---|
| SC-MARA-spline | 80.43 (10.88) | 0.95 (0.51) |
| MC-MARA-spline | 77.30 (13.98) | 0.53 (0.44) |
| SC-MARA-tPCA | 77.62 (7.39) | 0.95 (0.53) |
| MC-MARA-tPCA | 74.87 (10.03) | 0.57 (0.47) |
| HybridMARA-spline SG | 76.13 (7.38) | 1.28 (0.63) |
| HybridMARA-tPCA SG | 77.56 (7.02) | 1.34 (0.66) |
| TDDR | 91.28 (2.95) | 0.89 (0.46) |
| Wavelet | 79.67 (5.70) | 0.95 (0.39) |

### 3.3. Final PCA Step

Figure 6 shows the output signal of the previous BPF step (i.e., blue line) and the next PCA processing output (i.e., red line). On average, the four largest components of PCA were deleted according to 80% of the variance in the data. The average number of cancelled components was decreased to three in SC-MARA-Spline and MC-MARA-Spline. It is highly likely that the local spline corrections resulted in high power in the highest rank components including the large baseline drifts poorly reduced by this approach. Accordingly, an augmented power of cancelled components could be considered for such algorithms. Further, all algorithms displayed a similar and noticeable individual dispersion in the number of cancelled highest rank components, ranging from a minimum of two to a maximum of seven (IQR = 2 for MC/SC-MARA-spline, IQR = 4 for TDDR, IQR = 3 for other methods).

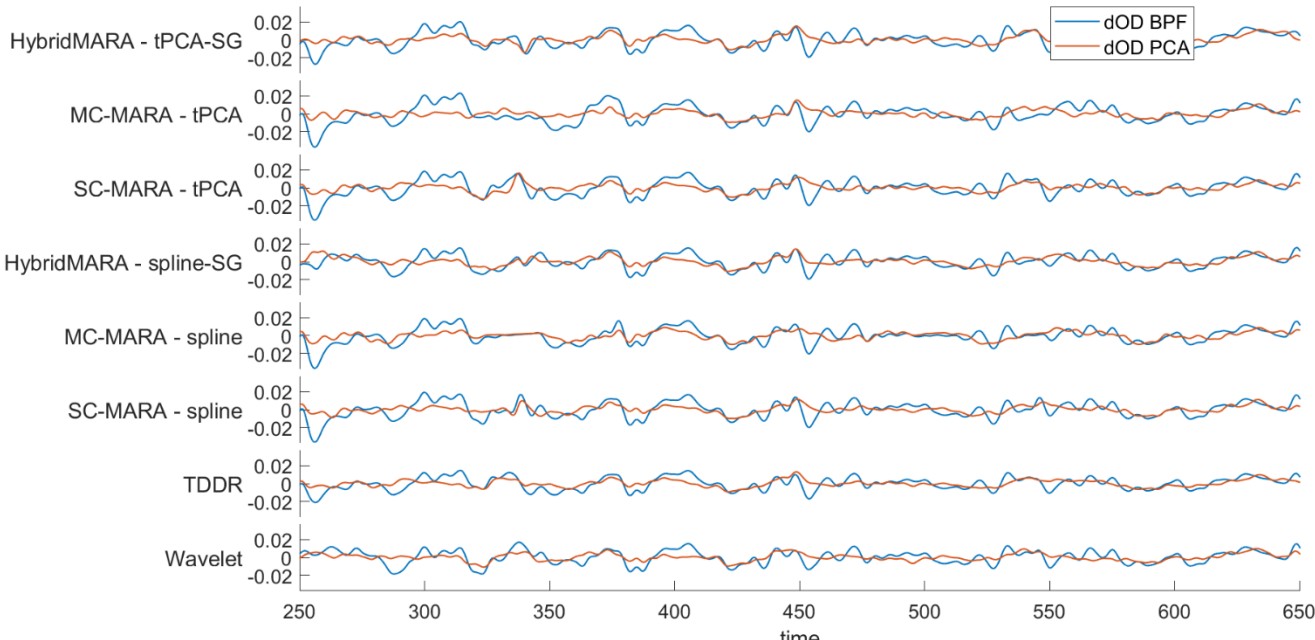

**Figure 6.** Single-channel graphical representation of the effect of PCA over $\Delta OD$ signals across pipelines. $\Delta OD$ signals are represented with respect to the BPF signals (blue line-input signal) and after PCA step (red line-output signal).

Despite the significant differences displayed in the single steps (MA, BPF, and PCA) the similarity of the final time courses of the eight pipelines (Figure 6, red lines) was remarkable. However, the results must be related to the statistical analysis for inferring activation vs. resting conditions and especially the overall localization of cortical activation as expected from the employed task. This is further evidenced by the final processing steps showing the HRF shapes and the significant group-level response maps, which are shown in Section 3.5.

### 3.4. Step and Overall ERD Assessment

Table 2 shows the percentage of ERD relevant to each method and each step: Table 2a) displays the percentage of ERD referred to in the specific step related to its input value; Table 2b) provides the cumulative percentage of the ERD effect shown as the rest energy compared to the unprocessed $\Delta OD$ signal (i.e., raw signal = 100%). The HYA group statistics, as the mean (std) across subjects and measurement channels, is reported. Table 2a clearly demonstrates the large differences in signal cleaning through single steps among the different methods. The highest impact in the MA reduction step was found in the global approaches of TDRR and Wavelet. A lower, yet large impact was found in Hybrid-MARA, with the lowest values in MARA. As for the last method, less impact was found in MC variants compared to the SC original algorithms. This result is counterintuitive, since the MC MA correction spanned larger time windows. However, this can be attributed to some baseline drift amplification due to local MARA corrections, which is less critical when longer windows are corrected after the MC MA detection logic. A significantly larger final power decrease was shown by both global methods: maximal and significant in TDRR; lesser yet noticeable in the wavelet method. Most likely, global methods, fixing a data-driven normal range, are also compelled to cut out the tails (i.e., out of the fixed 90 or 95 central percentiles) of standard components while canceling out the truly artifactual outliers. This eventually reduces the output signal power. However, it does not appear to negatively affect the final SNR.

**Table 2.** Summary table of the percentage Energy Relative Decrease (ERD) over optical density data along the pre-processing pipeline. Values are reported as mean (std) values across subjects and channels. (**a**) Progressive ERD at the single pre-processing step: unprocessed optical density ($\Delta OD_{raw}$), after MA removal ($\Delta OD_{MA}$), after BPF ($\Delta OD_{BPF}$) and finally after PCA ($\Delta OD_{PCA}$) as the final step of the processing pipeline. (**b**) Cumulative ERD with respect to the unprocessed optical density value (i.e., cumulative ERD = 100%).

| | (a) Single Step ERD [%] | | | (b) Cumulative ERD [%] | | |
|---|---|---|---|---|---|---|
| | $\Delta OD_{raw}$ to $\Delta OD_{MA}$ | $\Delta OD_{MA}$ to $\Delta OD_{BPF}$ | $\Delta OD_{BPF}$ to $\Delta OD_{PCA}$ | $\Delta OD_{MA}$ | $\Delta OD_{BPF}$ | $\Delta OD_{PCA}$ |
| SC-MARA-spline | −27.76 (17.12) | −73.91 (10.84) | −73.68 (18.44) | 72.23 (17.12) | 18.84 (8.92) | 5.79 (8.22) |
| MC-MARA-spline | −11.09 (26.09) | −80.30 (10.26) | −73.61 (18.39) | 88.90 (26.09) | 17.37 (9.64) | 5.37 (7.79) |
| SC-MARA-tPCA | −15.13 (23.40) | −76.64 (11.02) | −73.55 (18.40) | 84.86 (23.40) | 19.26 (8.85) | 5.91 (8.19) |
| MC-MARA-tPCA | −5.04 (31.95) | −79.14 (9.88) | −73.71 (18.46) | 94.95 (31.95) | 19.24 (8.31) | 5.79 (7.72) |
| HybridMARA-spline SG | −30.96 (23.67) | −73.01 (12.73) | −73.95 (18.49) | 69.03 (23.67) | 17.93 (9.29) | 5.58 (8.26) |
| HybridMARA-tPCA SG | −29.42 (18.06) | −72.83 (11.94) | −74.10 (18.52) | 70.57 (18.06) | 18.87 (9.12) | 5.77 (8.23) |
| TDDR | −60.28 (15.26) | −72.35 (9.10) | −73.71 (18.43) | 39.71 (15.26) | 10.88 (5.60) | 3.51 (5.60) |
| Wavelet | −39.79 (9.66) | −75.02 (10.63) | −74.04 (18.51) | 60.20 (9.66) | 15.48 (8.14) | 4.88 (7.77) |

Considering the last column of the cumulative percentage of ERD effects shown in Table 2b as the output of the whole preprocessing pipeline, two aspects are impressive: (i) the huge decrease of power finally obtained, which saves only about 5% of the raw signal; (ii) the modest differences found in the final outcomes of the eight pipelines, which further reassure us that essentially different MA approaches converge to similar outcomes, if followed by proper BPF and PCA steps. Conversely, large differences are found in

the partial ERD% outcomes described by Table 2a, which trivially reflects the different single-step impacts.

### 3.5. Block Average and Group-Level Results

An example of the block average in a representative subject (i.e., subject #4, which is the same one whose preprocessing steps were provided over Figures 4–6) is shown in Figure 7 relevant to left-hand motor task and a measurement channel placed over the right motor region (i.e., contralateral hemisphere to the hand performing the task).

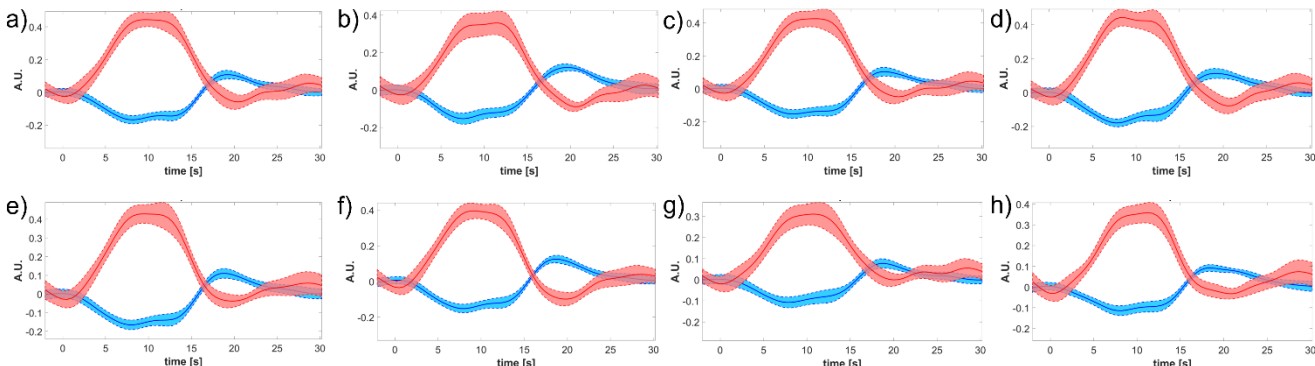

**Figure 7.** Subject-specific block-averaged responses across pipelines of $\Delta[HbO_2]$ and $\Delta[HbR]$ related to left-hand motor grasp and a measurement channel placed over the right motor region (i.e., contralateral hemisphere). (**a**) SC-MARA-spline, (**b**) MC-MARA-spline, (**c**) SC-MARA-tPCA, (**d**) MC-MARA-tPCA, (**e**) HybridMARA-spline SG, (**f**) HybridMARA-tPCA SG, (**g**) TDDR, and (**h**) Wavelet method.

As expected, a clear HRF is shown by the peak in $\Delta[HbO_2]$ (i.e., red curve) kept by the whole task duration and leading and trailing slope with the foreseen delayed dynamics. A concomitant drop in $\Delta[HbR]$ (i.e., blue curve) is also seen, but with a two-/three-fold scale reduction, which testifies to the regional cerebral blood flow increases. Dispersion of the 10 task repetitions is represented by the thickness of the respective lines (mean $\pm$ SE), which demonstrates a good SNR.

Interestingly, the whole cortex monitoring offered by the CW-fNIRS allows us to detect of secondary effects such as the inhibition of the ipsilateral (i.e., left) motor cortex, as if the activation of the left hand required to actively prevent the right hand from mirroring the left hand (Figure 8). Such specular inhibitions were already observed in other cortical areas, e.g., in the visual cortex under visual tasks [47,48] and push for further research under the hypothesis of being a core mechanism of brain organization and cerebrovascular coupling. It is worth remarking that the inhibitory HRF is consistently more than two-fold lower than the contralateral excitatory one (peak value in 0.3–0.4 A.U. range in Figure 7, 0.1–0.2 A.U. range in Figure 8). This effectively impacts the inhibitory $\Delta[HbR]$ amplitude, which is very small. Therefore, the fMRI BOLD signal, based on the sole $HbR$ and blind to $HbO_2$, might miss such inhibitory effects.

Further considering the example of subject #4 and the left motor cortex (i.e., Figure 7), Figure 9 displays the comparison between the block average response and the GLM fitting over the motor channel referred to Figure 7. The primary contralateral activation response is shown, which highlights the good superposition in the peak phase of $\Delta[HbO_2]$ and in the drop phase of $\Delta[HbR]$. However, some differences are displayed concerning the recovery phase, which is shorter in the GLM fitting due to the temporal trend of the canonical HRF (cHRF), which is derived from fMRI literature [49]. The latter displays broader plateaus centered around the block average positive $\Delta[HbO_2]$ and negative $\Delta[HbR]$ peaks. Such secondary shape differences are well explained by the GLM a-priori assuming a fixed standard HRF shape, which in turn is derived from the convolution of the cHRF with the on-off square wave of task onset-offset. Clearly, such an a priori regressor locks the

estimated shape and adapts only its amplitude. In this way, it retains details that are not detected in the single subject or may be smeared by the block average computation.

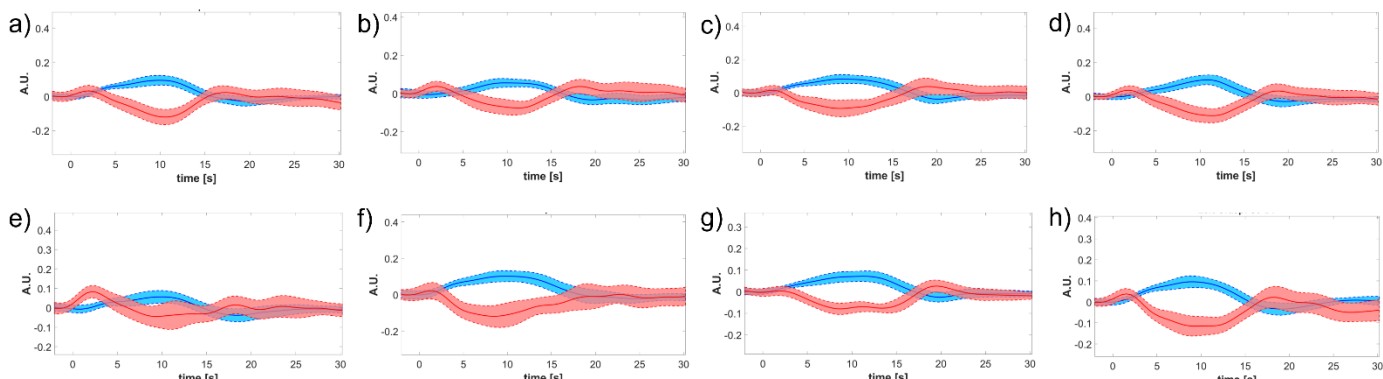

**Figure 8.** Subject-specific block-averaged responses across pipelines of $\Delta[HbO_2]$ and $\Delta[HbR]$ related to left-hand motor grasp and a measurement channel placed over the left motor region (i.e., ipsilateral hemisphere). (**a**) SC-MARA-spline, (**b**) MC-MARA-spline, (**c**) SC-MARA-tPCA, (**d**) MC-MARA-tPCA, (**e**) HybridMARA-spline SG, (**f**) HybridMARA-tPCA SG, (**g**) TDDR, and (**h**) Wavelet method.

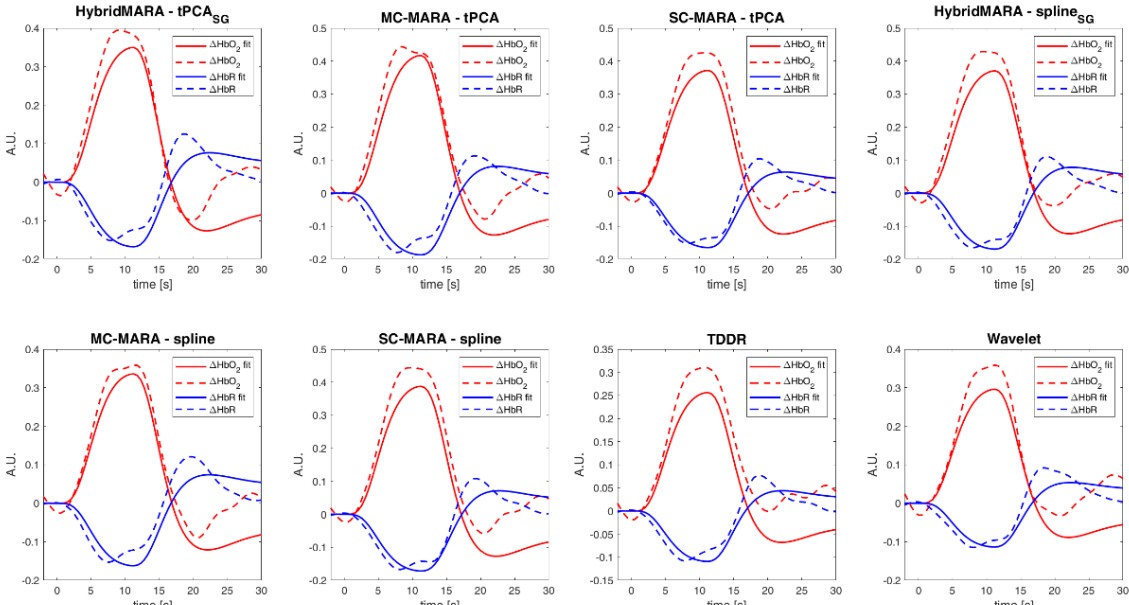

**Figure 9.** Visual comparison of subject-specific block-averaged response (dashed line) and GLM fitting (bold line) across pipelines. These results are related to left-hand motor grasp and a measurement channel placed over the right motor region (i.e., contralateral hemisphere) in accordance with Figure 7.

Importantly, all methods showed similar correspondences between the block-average and the GLM fitting. Nonetheless, the TDRR and Wavelet algorithms displayed slightly decreased response amplitudes (i.e., 0.4 A.U. vs. 0.3 A.U.), while keeping with the lesser power of their preprocessed signals. However, such a signal decrease was not accompanied by SNR drops.

Table 3 provides a numerical evaluation of the SNR of block averaged responses across subjects. Results refer to measurement channels placed over the motor region, following single subject representation of Figures 7 and 8. The SNR across task conditions over the contralateral motor region was within the 8.99–14.67 dB range for $\Delta[HbO_2]$ and 9.48–11.84 dB range for $\Delta[HbR]$, while ipsilateral activation presented reduced values in the 3.69–8.22 dB range for $\Delta[HbO_2]$ and 2.43–6.04 dB range for $\Delta[HbR]$. Overall, median

SNR values do not vary significantly across pipelines, confirming the graphical results provided in Figures 7 and 8. Among MARA approaches, SC-MARA often presents the highest median SNR value, both for tPCA and spline correction methods, while the lowest values are mostly attributed to the MC-MARA algorithm. In general, even lower SNR values are attributed to TDDR, since the decreased response amplitude compared to other algorithms reduces the total energy in the activation range of 6–12 s. Conversely, Wavelet methods provide SNR values more comparable to MARA methods, most probably due to a lower total energy in the baseline range of 24–30 s.

**Table 3.** Median value of block-averaged responses of SNR [dB] across all subjects, motor task conditions, and hemispheres. SNR was computed over motor channels displayed in Figures 7 and 8 and considered $t_{activation}$ in the 6–12 s range after stimulus onset, with $t_{baseline}$ in 24–30 s.

| | SNR Left Grasp Left Hemisphere [dB] | | SNR Left Grasp Right Hemisphere [dB] | | SNR Right Grasp Left Hemisphere [dB] | | SNR Right Grasp Right Hemisphere [dB] | |
|---|---|---|---|---|---|---|---|---|
| | $\Delta[HbO_2]$ | $\Delta[HbR]$ | $\Delta[HbO_2]$ | $\Delta[HbR]$ | $\Delta[HbO_2]$ | $\Delta[HbR]$ | $\Delta[HbO_2]$ | $\Delta[HbR]$ |
| SC-MARA-spline | 5.24 | 4.37 | 14.67 | 11.84 | 13.27 | 10.14 | 4.29 | 3.54 |
| MC-MARA-spline | 3.98 | 3.40 | 10.99 | 9.94 | 13.5 | 9.53 | 7.34 | 4.75 |
| SC-MARA-tPCA | 8.22 | 4.72 | 13.11 | 11.47 | 11.78 | 11.01 | 4.97 | 2.75 |
| MC-MARA-tPCA | 5.77 | 3.11 | 12.53 | 9.55 | 9.9 | 9.48 | 5.18 | 5.08 |
| Hybrid MARA-spline SG | 5.41 | 5.15 | 11.54 | 9.92 | 12.29 | 11.12 | 4.16 | 3.66 |
| Hybrid MARA-tPCA SG | 5.35 | 4.35 | 11.16 | 10.01 | 10.93 | 9.97 | 6.4 | 6.04 |
| TDDR | 4.52 | 2.43 | 8.99 | 11.15 | 9.17 | 9.60 | 3.69 | 3.50 |
| Wavelet | 5.39 | 3.52 | 11.51 | 9.62 | 13.77 | 10.94 | 6.63 | 5.19 |

The lack of substantial differences among different pipelines in the final single step and cumulative ERD, block-averaged responses, and SNR over the motor area is mainly related to the correspondence between the functional tasks and expected elicited brain areas. The temporal course and statistical significance were not affected. Notwithstanding, we suggest adopting preprocessing algorithms for MA reduction that present a channel-wise approach, such as SC-MARA, HybridMARA, TDDR, and Wavelet methods to prevent invasive interpolation over MA tracts which could lead to the loss of valuable signals if occurring at high rates. Among them, we suggest adopting the TDDR and Wavelet methods to increase the reproducibility since fewer parameters need to be set, supporting non-expert users in the signal processing.

For a better quantification of the above comparisons, Pearson's correlation and RMSE were computed between the block average and the GLM fitting curve in each subject and measurement channels, adapting the proposed approach by von Lühmann et al. [50]. All methods showed a significant linear correlation and Pearson's coefficient *r* across task and chromophores above the 0.6 (median in 0.613–0.682 range, IQR in the 0.287–0.328 range). Conversely, RMSE values associated with significant *r*-values were higher for $\Delta[HbO_2]$ (median in 0.022–0.031 A.U. range, IQR in 0.020–0.030 A.U. range) than $\Delta[HbR]$ (median in 0.009–0.014 A.U. range, IQR in 0.008–0.012 A.U. range). The reduced RMSE values in $\Delta[HbR]$ compared to $\Delta[HbO_2]$ could be attributed to the amplitude inequality between chromophores of block-averaged responses (Figures 7 and 9).

Passing to the whole HYA group analysis (alias, second level) of activation vs. resting condition significance maps, all methods were able to localize the activated contralateral motor region and the ipsilateral inhibition, as shown in Figures 10 and 11 for $\Delta[HbO_2]$ and $\Delta[HbR]$, respectively. The shown maps report significant *t*-values corrected for multiple comparison according to the Benjamini–Hochberg False Discovery Rate ($p_{FDR} < 0.05$). The *t*-test compared the average signal difference between the task and rest to the noise level at rest. Positive *t*-values (i.e., red color scale) indicate activation sites, while negative values (i.e., blue color scale) indicate inhibition. However, changing patterns were seen relevant to

other regions, such as secondary motor areas, frontal cognitive areas, and sensory, parietal, and occipital ones. Clearly, further research is needed to extend detection robustness and reliability to ancillary areas, which play a central role in the study of neuro-plasticity mechanisms addressed in neurological clinics.

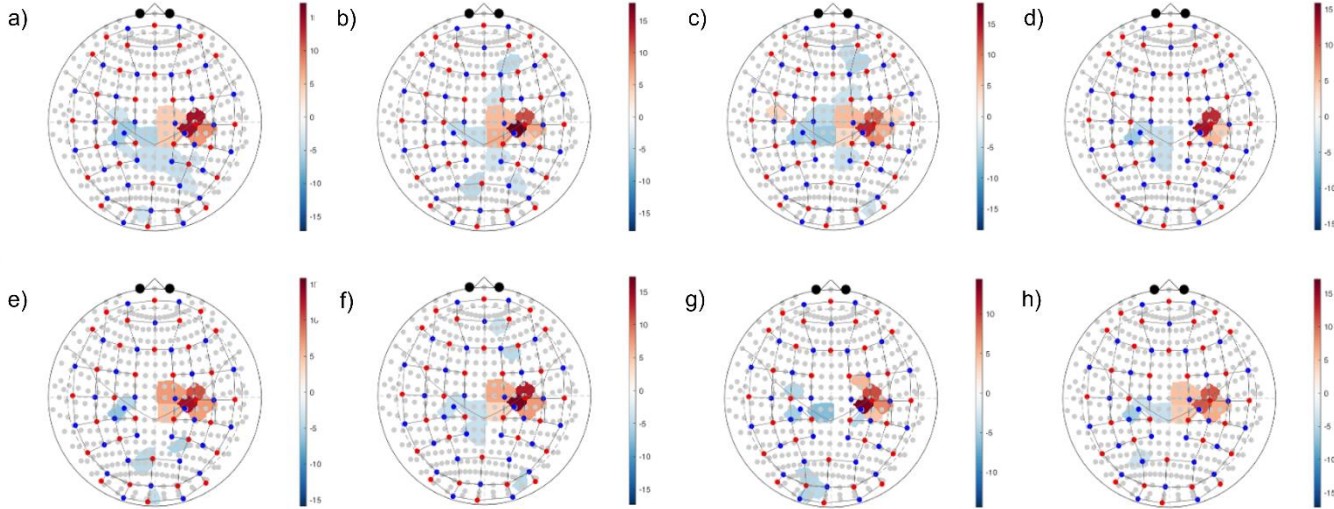

**Figure 10.** Second-level activation vs. resting condition statistical maps related to the left-hand motor grasp of $\Delta[HbO_2]$. Colorbars indicate significant t-values corrected for multiple comparisons ($p_{FDR} < 0.05$). Positive t-values indicate locations where the mean of $\Delta[HbO_2]$ activation was significantly higher than that in the resting condition, and conversely for negative values. (**a**) SC-MARA-spline, (**b**) MC-MARA-spline, (**c**) SC-MARA-tPCA, (**d**) MC-MARA-tPCA, (**e**) HybridMARA-spline SG, (**f**) HybridMARA-tPCA SG, (**g**) TDDR, and (**h**) Wavelet method.

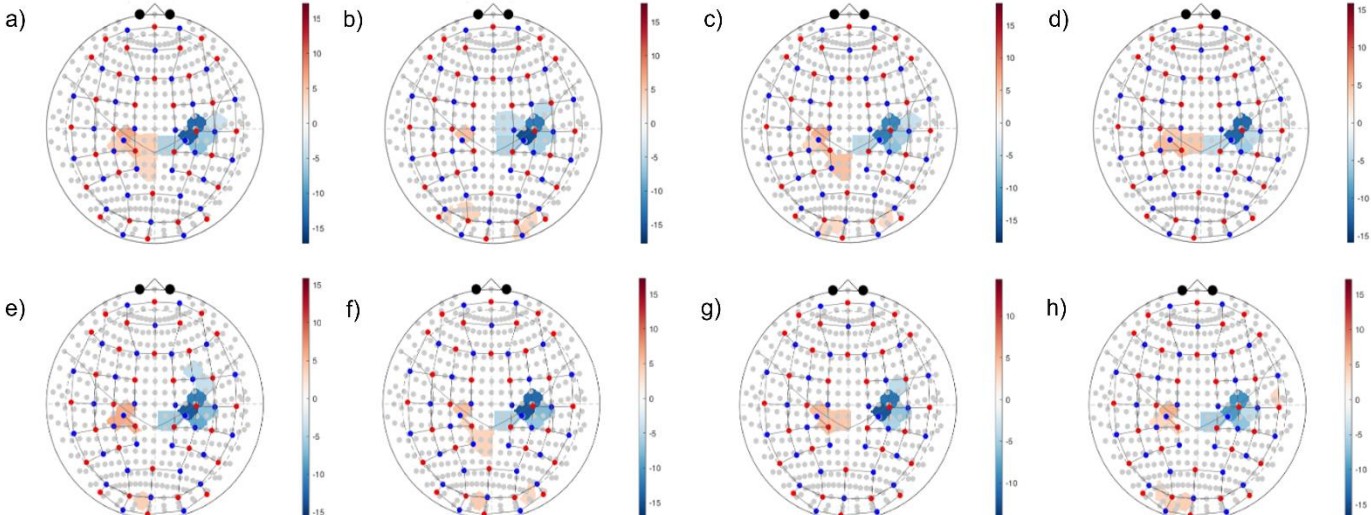

**Figure 11.** Second-level activation vs. resting condition statistical maps related to the left-hand motor grasp of $\Delta[HbR]$. Colorbars indicate significant *t*-values corrected for multiple comparisons ($p_{FDR} < 0.05$). Positive t-values indicate locations where the mean of $\Delta[HbR]$ activation was significantly higher than that in the resting condition, and conversely for negative values. (**a**) SC-MARA-spline, (**b**) MC-MARA-spline, (**c**) SC-MARA-tPCA, (**d**) MC-MARA-tPCA, (**e**) HybridMARA-spline SG, (**f**) HybridMARA-tPCA SG, (**g**) TDDR, and (**h**) Wavelet method.

## 4. Discussion

In this study, we proposed a comparison of the most used artifact reduction strategies in light of the common absence of separate artifact sensing systems using a commercial

CW-fNIRS instrumentation. In fact, Frequency or Time Domain fNIRS are still seldom used in clinical procedures because of their higher costs and complexity but indicate a promising method for developing more robust and reliable optical brain imaging methods [14,15,51]. Technological developments in optical imaging are moving towards the promotion of High Density Diffuse Optical Tomography (HD-DOT) techniques, which can cope with spatial resolution problems of traditional fNIRS thanks to overlapping measurement channels. HD-DOT allows us to have comparable localization of functional activation to fMRI, and it has already been validated in retinopy, motor, and language paradigms [52]. However, its clinical applications are still confined to specific fields, such as breast, muscle and fluorescence imaging [53], epilepsy, and neonatal/infant brain injuries [52], with very few exceptions in clinical neurology such as Parkinson's Disease [54].

Although CW-fNIRS is still not suitable as a stand-alone diagnostic device, its increasing use in different clinical and experimental domains requires that the technical issues, such as MAs and PCFs, must come after patients' tolerability and the opportunity to perform multiple acquisitions in a clinical rehabilitation context.

Moreover, another CW-fNIRS advantage is the possibility of having a complete brain cortex functional mapping, which is crucial when the study's objective is to monitor the brain plasticity mechanism due to rehabilitation programs [55–57]. Furthermore, for these purposes, the focus is on differential changes in $\Delta[HbO_2]$ and $\Delta[HbR]$ instead of quantification of these chromophores.

Nowadays, MA and PCF removal is provided by integrating auxiliary measurements such as short separation channels that have the exceptional advantage of regressing out scalp hemodynamics in first- and second-level analyses [16,17,58]. This approach can be integrated with other physiological signals such as blood pressure, movement/accelerometer, respiration, and photo plethysmography measurements [18].

However, these auxiliary measurements are not always available with commercial systems or can be integrated within the experimental clinical set-up. Therefore, a systematic evaluation of employed processing pipelines and related algorithms' effect on optical signals must be carefully considered. Indeed, as shown from our results, the impact on the statistical significance and especially the localization of cortical activation can be greatly affected. This is a severe issue for translating this technique in clinical neuroimaging. When performing an assessment protocol in the context of a rehabilitation program, the activation of additional cortical areas as a supporting brain source to execute a task is the actual matter of research [59].

Emerging from our results (Section 3.5), the block-average response over the motor areas and the respective SNR across all subjects did not produce substantial differences among the different pipelines. This effect is mainly related to the correspondence between the functional task (i.e., motor grasping task) and expected elicited brain areas. In addition, these limited differences are mainly related to amplitude variations, which are not the principal purpose of the CW-fNIRS. Conversely, addressed features such as the temporal course, SNR, and statistical significance are seemingly not affected. Nevertheless, we suggest adopting preprocessing algorithms for MA reduction that present a channel-wise approach, such as SC-MARA, HybridMARA, TDDR, and Wavelet methods. Indeed, MC variants exert a more invasive interpolation over MA tracts, which may eliminate useful signals if occurring at high rates throughout the whole recording.

SC-MARA and HybridMARA methods provide a higher SNR of block-averaged responses compared to TDDR and Wavelet over cerebral areas where we expect to have a significant activation. However, the TDDR and Wavelet methods have the advantage of requiring fewer input parameters and no adoption of user-defined thresholds compared with the SC-MARA and HybridMARA methods, hence promoting reproducibility of results. Therefore, when adopting SC-MARA and HybridMARA methods, which have comparable metrics to TDDR and Wavelet, researchers must clearly state their experimental set-up and adopted parameters [29]. The central issue concerning the adoption of different processing algorithms, especially for MAs, arises when inferring cortical activation over

supplementary areas, here represented by non-motor areas. Indeed, while for HYA we could expect a correspondence between the task proposed and the activated areas, when pathological conditions occur, such a coherence is not assured and even not predictable.

Another fascinating insight that emerged from our results is related to the ipsilateral hemodynamic response observed during the hand grasping. In ipsilateral motor areas, $HbR$ remained stable at its baseline level, whereas the $HbO_2$ level decreased (Figure 8). This phenomenon has already been observed in fMRI as Negative BOLD Responses (NBRs), but it is still debated and not fully understood, making such a mechanism not always directly interpretable. Recent studies have shown that multimodal integration could provide a complete perspective on this phenomenon [60]. Optical techniques could lead to a more comprehensive understanding of NBRs, even if such an approach is still limited in its methodological application [61].

This study presents some limitations that could be addressed with future methodological assessments. Namely, a further refined methodological approach will necessarily require the assessment of these algorithms through simulation of HRF superimposed on resting-state data [34]. A statistical comparison of activation maps concerning regression with short-separation channels will also give additional insights into the actual imbalance between proposed algorithms.

## 5. Conclusions

In conclusion, current fNIRS research is still far from an objective standardization of pre-processing and analysis pipelines [12,61] due to the heterogeneity of employed commercial and custom instrumentation. However, we demonstrated that by adopting a channel-wise approach in signal processing it is possible to obtain specific information regarding the impact of each algorithm on the estimation of functional activation, either at the statistical level or mean response level.

**Author Contributions:** A.B., F.S.I., G.B. and F.B. conceptualized the study. F.B. acquired the financial support for the project leading to this publication. G.B. and F.B. conceived the methodological design. F.S.I. and A.B. conducted the research and investigation process, specifically performing the experiments and data collection; G.B. and F.B. were involved in work supervision. A.B. implemented the computer code and supporting algorithms. A.B. and F.S.I. wrote the manuscript in consultation with G.B. and F.B. All authors have read and agreed to the published version of the manuscript.

**Funding:** This study was co-funded by the Lombardy Region (Announcement POR-FESR 2014-2020) within the project named "Sistema Integrato DomiciliarE e Riabilitazione Assistita al Benessere" (SIDERA^B).

**Informed Consent Statement:** Informed consent was obtained from all subjects involved in the study.

**Conflicts of Interest:** The authors declare no conflict of interest. The funders had no role in the design of the study; in the collection, analyses, or interpretation of data; in the writing of the manuscript, or in the decision to publish the results.

## Appendix A

### *Appendix A.1. Single-Channel MARA with Spline Correction (SC-MARA-Spline)*

SC-MARA (Movement Artifact Reduction Algorithm) with spline correction is the original algorithm version that performs MA reduction by separating identification and correction phases [21]. MA detection is based on the computation of the moving standard

deviation of $\Delta OD(t, \lambda)$ signals according to an odd sliding window of $w = 2k + 1$ samples large enough to contain MA features:

$$mSTD(t) = \frac{1}{2k+1} \sqrt{\sum_{j=-k}^{k} \Delta OD(t+j, \lambda) - \frac{1}{2k+1} \left( \sum_{j=-k}^{k} \Delta OD(t+j, \lambda) \right)^2} \quad \text{(A1)}$$

In the present work, we set the moving sliding window to five samples, which almost correspond to epochs of three seconds according to our experimental set-up (details are presented in Section 2.8).

Since MAs are mainly associated with higher signal variations compared to typical hemodynamic oscillations, the resulting $mSTD(t)$ signal allows us to detect epochs of the signals affected by MAs. Hence, MA windows are defined by setting a user-defined threshold over the resulting $mSTD(t)$ signal. Namely, in this work we set a threshold of 5, which indicates that a portion of the signal is labelled by MA if exceeding $5 \cdot mSTD(t)$ within the considered window length of three seconds. Additionally, other methods can be implemented to automatically define a threshold. Among them we cite the Triangle method for unimodal histograms [62]. In brief, the choice of the threshold is given by computing the maximum distance of the perpendicular segment to the line connecting the first and last bin of $mSTD(t)$ histogram. Thus, MA are identified over time periods where $mSTD(t)$ is above the computed threshold.

Finally, the MA-labelled windows are corrected by subtracting the anomalous baseline fitted via spline interpolation. The spline parameter $p$ plays the major role: $p = 1$ would spline through all samples thus cancelling all dynamics since corresponding to the natural cubic spline interpolant; $p = 0$ would implement a linear detrend since implementing least-square straight-line fitting. We followed the recommendation in the original work setting $p = 0.99$, which identifies the trend to be subtracted, even in the presence of high discontinuities, leaving the informative dynamics around it.

### Appendix A.2. Multi-Channel MARA with Spline Correction (MC-MARA-Spline)

The MC-MARA-Spline variant we experimented was identical in terms of correction step to SC-MARA-Spline in the starting SC detection of MAs. However, the identification step of MA windows was firstly applied as SC-MARA-Spline and next extended to all channels according to OR logical operator (i.e., if an MA window is detected over a single channel, it is extended to all epochs across other channels). Therefore, the major difference with its analogous SC-variant is that many channels apparently within the normal range underwent correction due to the detection of a MA in at least another channel. Both the MC- and SC-MARA-Spline algorithms were implemented through *hmrMotionArtifactByChannel* and a revised version of *hmrMotionCorrectSpline* functions of Homer2 developer's version.

### Appendix A.3. Single-Channel MARA with Targeted PCA Correction (SC-MARA-tPCA)

We also experimented with the combination of the SC-MARA detection by employing the targeted PCA (tPCA) method proposed by Yücel et al. [23] at the correction step. This method is based on the hypothesis that MAs globally represent the most prominent variation of measured optical signals. Hence, larger components obtained by PCA decomposition performed over the identified MA windows are cancelled out. Namely, larger principal components explain the higher percentage in the variance of the data. Therefore, they are sorted in decreasing order and the signal not associated with them is back-projected as motion-corrected signal. Such time targeting intends to overcome the limitations of the overall PCA, which hardly captures the non-stationary features of MAs. The critical parameter in the correction step is the power reduction imposed by the cancellation of the largest tPCA components. In line with the original work, we set this method to remove up to 97% of the variance in the data.

*Appendix A.4. Multi-Channel MARA with Targeted PCA Correction (MC-MARA-tPCA)*

The respective MC-MARA-tPCA variant of the previous algorithms implements the same SC-MARA detection criterion but extends MA labelling to all channels according to OR logical operator. Accordingly, labelled MA tracts (now common to all channels) undergo tPCA correction. Interestingly, this approach can be said MC both in MA labelling and in the correction step. At first sight, this might be sensible. However, if the labelled tracts tend to the whole recording duration, time targeting would be lost, thus reverting to the limitations of the overall PCA. Both the MC- and SC-MARA-tPCA algorithms were implemented through *hmrMotionArtifactByChannel* and a revised version of *hmrMotionCorrectPCA_Ch* functions of Homer2 developer's version.

*Appendix A.5. Hybrid MARA with Spline Correction and Savitzky-Golay Filtering (HybridMARA-Spline SG)*

This algorithm implements the strategy proposed by Jahani et al. [22], which integrates the previous MARA identification based on $mSTD(t)$ to detect baseline shifts and spike artifacts. These two instances are then separately detected and corrected, hence the "Hybrid" name given to the approach. In brief, MA identification is provided by applying both the MARA method (as detailed in Appendix **??**), since it is more suited for baseline shifts identification, and an additional convolution between the lowpass-filtered optical signal and a Sobel derivative kernel (coefficients $[-1\ 0\ 1]$), which is conversely more prone to deal with spike artifacts. A threshold over the resulting signals is computed according to the union of interquartile statistics, detecting outliers if falling outside $[Q1 - 1.5 \cdot IQR; Q3 + 1.5 \cdot IQR]$, where $Q1$ and $Q3$ are the respective first and third quartile. Hence, baseline shifts are identified by computing the maximum amplitude variation in the motion-free part of the signal according to a sliding window of 0.5 s (i.e., amplitude variations higher than half of heartbeat oscillations) and corrected through spline interpolation if $SNR > 3$. Finally, the resulting signals, remaining spike artifacts, and motion-free part of the signals with $SNR < 3$ are smoothed out by Savitzky–Golay (SG) filtering, which is a digital polynomial filter that substitutes a sample by the trend fitting the adjacent ones. This work employs a 3rd-order SG filter according to window lengths of 3 s, following MARA approaches and being shorter that HRF dynamics. This HybridMARA-tPCASG approach was implemented through a revised version of the *hmrMotionCorrectSplineSG* function of Homer2 developer's version.

*Appendix A.6. Hybrid MARA with Targeted PCA and Savitzky-Golay Filtering (HybridMARA-tPCA SG)*

In line with the approaches proposed in the work of Jahani et al. [22] and MARA approaches, we also tested tPCA correction. This method implements the same identification and SG smoothing strategy as described in Appendix A.5. Conversely, the correction of baseline shifts by spline interpolation is indeed substituted by the tPCA method. In general, performing baseline shift correction with tPCA reduces the possibility of removing physiological oscillations over motion-contaminated portions of the signal. Additional motivations regarding the motivation for tPCA correction are referenced in Appendix A.3. This HybridMARA-tPCASG approach was implemented through a revised version of *hmrMotionCorrectPCASG* function of Homer2 developer's version.

*Appendix A.7. Temporal Derivative Distribution Repair (TDDR)*

This strategy has been recently proposed by Fishburn et al. [24] to iteratively address both MA trends and spikes under the assumption that the derivative signal can be approximated through a Gaussian distribution whose outliers are MA fluctuations. Indeed, the signal derivative is computed as the unsmoothed increment between adjacent samples $y'^{(t)} = \Delta OD(t) - \Delta OD(t-1)$, and outliers are attenuated or cancelled by the iter-

ative estimation of instantaneous weights $w(t)$, which follow Tukey's bi-weight function $w(t) = \left(1 - d^2(t)\right)^2$

$$w(t) = \begin{cases} \left(1 - d(t)^2\right)^2, & |d(t)| < 1 \\ 0, & otherwise \end{cases} \tag{A2}$$

The instrumental variable *d(t)* rescales the absolute deviations from the weighted mean $\mu$ to include the 95% of values in the $|d(t)| < 1$ range, under the hypothesis of Gaussian distribution

$$d(t) = \frac{|y'(t) - \mu|}{\sigma \cdot 1.4685} \tag{A3}$$

$$\mu = \frac{\sum_{i=1}^{T} w(t_i) y'(t_i)}{\sum_{i=1}^{T} w(t_i)} \tag{A4}$$

Importantly, the instrumental variable is based on the standard deviation of residuals $\sigma$, which overlooks the outliers to be next cancelled via a classical conversion from the median of absolute values. The actual correction takes place only after iteration convergence by robust weighting to centered derivative $y'(t) = w(t)[y'(t) - \mu]$. Finally, the corrected derivative signal is integrated back $x'(t) = \sum_{1}^{t} y'(t)$ and centered with respect to the mean of the uncorrected signal. We employed the methods provided in the NIRS Brain AnalyzIR Toolbox [34].

### *Appendix A.8. Wavelet-Based Detection and Correction (Wavelet)*

Molavi and Dumont [25] proposed their wavelet bases strategy mainly addressing spiking MA removal. Nonetheless, the large range of wavelet scales considered can also address baseline shifts and drifts. The algorithm starts with a discrete wavelet transform (DWT) and employs Daubechies 2 (db2) wavelet at four different levels. The distribution of wavelet coefficients within each scale is considered approximately Gaussian, with zero mean and standard deviation robustly evaluated from the median of absolute values. Hence, the cutoff, which is suggested to be set outside the 90% range, is not influenced by the outliers to be cancelled, which permits a degree of cleaning proportional to the artifact rate. To avoid effects due to the shift sensitivity of the DWT, cleaning is performed on all possible circular shifts, whose results are next realigned and averaged. In this work, outlier identification was based on the wavelet coefficients exceeding 1.5 times the interquartile range according to the *hmrMotionCorrectWavelet* function of Homer2 developer's version.

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
