# Peer review of "Assessment of fNIRS Signal Processing Pipelines: Towards Clinical Applications"

_applsci, doi:10.3390/app12010316_

Round 1
Reviewer 1 Report
ABSTRACT:
lines 11-12: I would point out additional or different pitfalls of the fMRI as fNRIS has for sure the same drawbackes mentioned: poorly cooperating subjects and motion artifacts.
line 26: down up to 4% energy... I would recommend to use a differend wording for better understanding
line 29: inhibitions in the ipsilateral one... I would not mention that in the abstract at this fact has only been shown for one single subject (out of 23) - how did the signals look like in within the rest of the group?
INTRODUCTION ++
page 1, line 44: I would not put it as "a massive overreaction of perfusion"
page 2, lines 48, 49 sentences start both with "In turn,..." also in lines 66 "due" twice, line 88: As a resultS (no s), page 3 line 102 is there and "and" missing or was the "may" not deleted, page 4, line 86 equation IS designed... - these are just examples given, so I would really recommend to proof read the paper again, as a lot of spelling mistakes, missing semicolons (eg. references), word repetions and unnecessary abbreviations (eg.HYA healthy young adults) are used and makes it for the reader a bit cumbersome to follow arguments.
page 2, line 72 again (and this is to be taken into account in all explanations/arguments: a non-cooperating subject/participant is a pitfall for ANY study, not only for the fMRI.
MATERIALS + METHODS
Overall a question from my site: maybe all forms used within this sections could be added as supplementary, because there are no novel mathematical calculations but application of well established algorithms.
page 5 line 209: I assume there is a multiplication sign missing?
How many of the 23 participants were excluded in total? (eg. too high movement artifacts or too low SNR during experiments?
page 10 line 438: lesser margin - either less margin or lower margin
page 11 line 481: no explantion of APD (avalanche photodiode)
page 11 lines 490 ff: why was the block design chosen not to be symmetric, i.e. 10 sec. activity and 20 sec. resting/non activity, instead of 20:20 seconds?
What were the geometrical/spatial distances of the probes on the heads?
page 13 line 550: are the given percentages correct or only in the wrong order "1,10%-0,90%", next line its "0,32%-0,42%" and in line 557 its "6,46%-5,85%"
As there have been calculated comparisons of 8 different artifact reduction strategies I don't see the benefit of showing Figure 3, as only the 3 MARAs are shown and depict in the corrected percentage quite a lot of differences in the means but also in std.
page 16, lines 645, 646 clearly demonstrates the large differences - I for sure do agree, also when having a look at the numbers from deltaODraw to deltaODma - but I would appreciate, if discussed in the results a bit more in detail wich algorithm enables to get the best and most repeatable results as an advice for other users of fNRIS.
FIGURES:
Overall, if printed out, the Figures 4, 5 and 6 are impossible for the reader to visually identify parameters.
And for clarity reasons I would recommend to keep colors for the same calculated/depicted signals: eg. Fig. 4 dOD motion corrected is reddish, and in Fig. 5 it switches to blue color and the reddish one is now the bandpass filterd signals.
a.u. is not depicted everywhere (eg. missing in Fig. 7)
pages 18-24: the presented results from are for sure interesting regarding the inhibitory HRF within the contralateral hemisphere - however, this is only one single subject (as already mentioned above) - it should be at least discussed how the other participants showed activity, it can also happen that there is co-activation on both sides of the hemispheres (motoric region) even though one-handed movement pattern only.
Question out of interest: was there no activaion of the visual cortex at all? I assume the participants were looking at a display during the task and sometimes there is a higher activation of the visual cortex during the tasks, so at group comparison level there might be still some signals left.
SUM:
I think its clearly pointed out what the aim of the paper was and how the procedure was implemented and results were received.
Drawback: missing explanations, how other participants activations looked like and no novelty regarding new set-up or calculations.
Overall the paper is written well regarding details and the pipeline of the calculations, however, very much details and abbriviations are used, which might distract the reader - I would recommend to proof read the paper regarding english phrasing to ensure a more clear structure, as this would improve the intelligibility of this paper significantly.
Kind regards
Author Response
According to reviewer concern, we highlighted the revisions in the manuscript according to “track changes” option and we addressed the specific questions point by point in the following paragraphs.
ABSTRACT
[RW1 - point] lines 11-12: I would point out additional or different pitfalls of the fMRI as fNRIS has for sure the same drawbacks mentioned: poorly cooperating subjects and motion artifacts.
[Response] Surely, fNIRS presents the same limitations as fMRI with respect to the poorly cooperating subjects. However, people not eligible for MRI, either due to practical limitations (i.e., presence of metal implants) or due to suffering from claustrophobia could be eligible for fNIRS. We added these modifications in the at line 12 – 15.
[RW1 - point] line 26: down up to 4% energy... I would recommend to use a differend wording for better understanding
[Response] This line was changed in “The mean ERD% of pre-processed signals with respect to initial raw signal energy was up to 4%”.
[RW1 - point] line 29: inhibitions in the ipsilateral one... I would not mention that in the abstract at this fact has only been shown for one single subject (out of 23) - how did the signals look like in within the rest of the group?
[Response] Ipsilateral inhibition was shown over group-level statistical maps. Hence, single subject trends are taken as indication of group-level statistics. However, following your suggestion we removed this sentence from the abstract.
INTRODUCTION
[RW1 - point] page 1, line 44: I would not put it as "a massive overreaction of perfusion"
[Response] We change the wording as follow: “is the critical variation of perfusion” at line 47 page 2.
[RW1 - point] page 2, lines 48, 49 sentences start both with "In turn,..." also in lines 66 "due" twice, line 88: As a resultS (no s), page 3 line 102 is there and "and" missing or was the "may" not deleted, page 4, line 86 equation IS designed... - these are just examples given, so I would really recommend to proof read the paper again, as a lot of spelling mistakes, missing semicolons (eg. references), word repetions and unnecessary abbreviations (eg.HYA healthy young adults) are used and makes it for the reader a bit cumbersome to follow arguments.
[Response] We state that all the line-specific revisions concerning this aspect can be found across the text. Unfortunately, the correspondence to your revisions cannot be directly addressed due to the “track changes” option.
[RW1 - point] page 2, line 72 again (and this is to be taken into account in all explanations/arguments: a non-cooperating subject/participant is a pitfall for ANY study, not only for the fMRI.
[Response] In line with the statement in the Abstract section, our intent to define non-cooperating subjects is referred to people not eligible for a MRI scan, either due to practical limitations (i.e., presence of metal implants) or due to suffering from claustrophobia. A more precise statement concerning this aspect is provided at line 79 – 82. We have revised the entire manuscript in this perspective.
MATERIALS + METHODS
[RW1 - point] Overall a question from my site: maybe all forms used within this sections could be added as supplementary, because there are no novel mathematical calculations but application of well-established algorithms.
[Response] Sections from 2.5.1 to 2.5.8 concerning the description of motion artifact algorithms were moved to Appendix A as suggested, specifying this modification within Section 2.5.
[RW1 - point] page 5 line 209: I assume there is a multiplication sign missing?
[Response] We added the missing multiplication sign in Eq. 5.
[RW1 - point] How many of the 23 participants were excluded in total? (eg. too high movement artifacts or too low SNR during experiments?
[Response] A statement regarding the number of excluded participants has been introduced from line 282 to 285. Overall, 5 out of 23 participants were excluded according to the exclusion criterion presented in Section 2.4.
[RW1 - point] page 10 line 438: lesser margin - either less margin or lower margin
[Response] We confirmed that your modification with “lower margin” is correct and we modified that coherently.
[RW1 - point] page 11 line 481: no explantion of APD (avalanche photodiode)
[Response] The fNIRS device employed in this study together with all available technologies (i.e. LED for sources and APD for detectors) are commercial systems. All details regarding source and detector components are available on the manufacturer site (https://nirx.net/nirscout).
[RW1 - point] page 11 lines 490 ff: why was the block design chosen not to besymmetric, i.e. 10 sec. activity and 20 sec. resting/non activity, instead of 20:20 seconds?
[Resposnse] The chosen block design is not symmetric for two main reasons: [1] the employed block design repetition frequency does not filter out the HRF task-driven oscillations, while having a repetition frequency of will get closely to this lower margin, [2] hand-grasping and similar other experimental designs such as finger-tapping found in literature have been shown to require a 2s up to 10s task duration to provide an effective hemodynamic response (Kashou et al., Neurophotonics, 3(2), 025006 (2016)).
[RW1 - point] What were the geometrical/spatial distances of the probes on the heads?
[Response] Mean source-detector distances, as referred to the virtual co-registration of this configuration over the Colin27 atlas, was 3.46 cm. As well, the positioning of probes followed 10-10 electrode system for EEG as displayed in Figure 1.
[RW1 - point] page 13 line 550: are the given percentages correct or only in the wrong order "1,10%-0,90%", next line its "0,32%-0,42%" and in line 557 its "6,46%-5,85%"
[Response] Provided values are in the correct order and indicate the mean percentage - across subjects and measurement channels - of portions of the signal identified as motion artifact with respect to its whole length. The percentage of 1,10%-0,90% (median - IQR) is referred to HybridMARA, 0,32%-0,42% (median - IQR) to SC-MARA and 6,46%-5,85% (median - IQR) to the tested MC-MARA version.
[RW1 - point] As there have been calculated comparisons of 8 different artifact reduction strategies I don't see the benefit of showing Figure 3, as only the 3 MARAs are shown and depict in the corrected percentage quite a lot of differences in the means but also in std.
[Response] Figure 3 provides a comparison of motion artifact length and percentage referred to only three methods, even if 8 different algorithms were tested. These boxplots are referred to the identification step of HybridMARA, MC- and SC-MARA versions. Therefore, the results do not vary between HybridMARA-tPCA SG vs. HybridMARA-splineSG, SC-MARA-tPCA vs. SC-MARA-spline, MC-MARA-tPCA vs. MC-MARA-spline algorithms. Additionally, results are not reported for Wavelet and TDDR methods since these algorithms perform identification and correction steps simultaneously and do change the overall trend of the signal. Therefore, computing motion artifact length and percentage over processed signals will give results close to 100% and the overall signal length. For these reasons, a statement to clarify this aspect was added in Section 3.1 line 574 – 582.
[RW1 - point] page 16, lines 645, 646 clearly demonstrates the large differences - I for sure do agree, also when having a look at the numbers from deltaODraw to deltaODma - but I would appreciate, if discussed in the results a bit more in detail wich algorithm enables to get the best and most repeatable results as an advice for other users of fNRIS.
[Response] In line 652-661pag 18 we added “The lack of substantial differences among different pipelines in the final single step and cumulative ERD, block-averaged responses and SNR over the motor area are mainly related to the correspondence between the functional tasks and expected elicited brain areas. The temporal course and statistical significance are not affected. Notwithstanding, we suggest adopting preprocessing algorithms for MAs reduction that present a channel-wise approach, such as SC-MARA, HybridMARA, TDDR and Wavelet methods to prevent invasive interpolation over MAs tracts which could lead to the loss of valuable signal if it occurring at high rates. Among them, we suggest adopting TDDR and Wavelet methods to increase the reproducibility since demanding fewer parameters to be set, supporting non-expert users in the signal processing”.
FIGURES
[RW1 - point] Overall, if printed out, the Figures 4, 5 and 6 are impossible for the reader to visually identify parameters.
[Response] We have re-scaled the figures as much as possible to increase the visibility of labels and parameters.
[RW1 - point] And for clarity reasons I would recommend to keep colors for thesame calculated/depicted signals: eg. Fig. 4 dOD motion corrected is reddish, and in Fig. 5 it switches to blue color and the reddish one is now the bandpass filtered signals .a.u. is not depicted everywhere (eg. missing in Fig. 7)
[Response] Concerning Figure 4, Figure 5 and Figure 6, we followed this convention: blue lines do reflect the input signal of the considered processing step (i.e. motion artifact removal for Figure 4, bandpass filtering in Figure 5 and PCA in Figure 6), while the red lines do reflect the output signal processed signal. We highlighted this concept in the caption of each figure and, in line with the previous point, we have rescaled the legend to have a better visibility. Finally, we reported the A.U. label over each subplot of Figure 7 and Figure 8 as requested.
[RW1 - point] pages 18-24: the presented results from are for sure interesting regarding the inhibitory HRF within the contralateral hemisphere- however, this is only one single subject (as already mentioned above) - it should be at least discussed how the other participants showed activity, it can also happen that there is co-activation on both sides of the hemispheres (motoric region)even though one-handed movement pattern only.
[Response] For sure, further discussion regarding ispilateral inhibition over each subject would add further evidence. However, the group-level statistical maps in Section 3.5 provide evidence of this ipsilateral inhibition, hence reflecting possible considerations which could be addressed over single subjects. Moreover, we added in the manuscript a SNR measure of block averaged responses across all subjects and over bilateral motor channels. This measure considers the ratio between the total energy of the block averaged responses in [6 ; 12]s (i.e. activation) vs. [24 ; 30]s (i.e. baseline) after stimulus onset. The SNR metric is introduced in Methods at Section 2.8 at line 430 – 435, while related results and discussion are provided at line 633 – 646 and 750 – 725.
[RW1 - point] Question out of interest: was there no activation of the visual cortex at all? I assume the participants were looking at a display during the task and sometimes there is a higher activation of the visual cortex during the tasks, so at group comparison level there might be still some signals left.
[Response] The proposed grasping task recruits massively and slightly exclusively motor and cognitive resources; although significant activations over the visual cortex could arise because of the displayed stimulus on the screen, they would be hidden in the multiple comparisons due to the strong significance of the motor and frontal areas activations.
Reviewer 2 Report
An assessment of fNIRS signal processing pipelines is clearly an important topic, if fNIRS will ever be fully useable in any individual neurology patients' clinical applications.
While it demonstrated in some detail (albeit without decent editing), many ways to skin the fNIRS artifact cat, it did not compare them in the only way that matters.
That is, Signal to Noise Ratio (SNR). It touched on SNR briefly, but never gave numbers, and the plots in figure 7 are all nearly the same.
I do not know why the authors danced away from the SNR number.
The SNR issue would need to be addressed much more thoroughly, and edited for clarity and conclusiveness, to merit publication.
Author Response
[RW2- point] An assessment of fNIRS signal processing pipelines is clearly an important topic, if fNIRS will ever be fully useable in any individual neurology patients' clinical applications. While it demonstrated in some detail (albeit without decent editing), many ways to skin the fNIRS artifact cat, it did not compare them in the only way that matters. That is, Signal to Noise Ratio (SNR). It touched on SNR briefly,but never gave numbers, and the plots in figure 7 are all nearly the same. I do not know why the authors danced away from the SNRnumber. The SNR issue would need to be addressed much more thoroughly, and edited for clarity and conclusiveness, to merit publication.
[Response] According to reviewer concern, we agree that providing more information over the SNR of measured hemodynamic response is an important aspect for the assessment of the signal processing pipeline. Therefore, we added in the manuscript a SNR measure of block averaged responses across all subjects and over bilateral motor channels. We also added Table 3 to summarize median SNR values across pipelines, tasks (i.e., left vs. right hand grasping) and hemispheres.
We evaluated the SNR over the ratio between the total energy of the block averaged responses in [6 ; 12]s (i.e. activation) vs. [24 ; 30]s (i.e. baseline) after stimulus onset. The SNR metric is introduced in Methods at Section 2.8 at line 430 – 435, while related results and discussion are provided at line 633 – 646 and 750 – 725.
Median SNR values do not vary significantly across pipelines. Among MARA approaches, SC-MARA often presents the highest median SNR value, both for tPCA and spline correction methods, while lowest values are mostly attributed to MC-MARA algorithm. Even lower SNR values are attributed to TDDR and Wavelet methods, since they do provide decreased response amplitude compared to other algorithms. Nevertheless, Wavelet method provides SNR values more comparable to MARA methods.
Overall, SNR results support the main findings both at individual level (i.e., block average responses provided in Figure 7 and Figure 8) and over group-level activation maps (i.e., Figure 10 and Figure 11). We can conclude that SC-MARA and HybridMARA methods generally provide a higher SNR of block averaged responses compared to TDDR and Wavelet over cerebral areas where we expect to have a significant activation. However, TDDR and Wavelet methods have the advantage of requiring less input parameters and no adoption of user defined thresholds than SC-MARA and HybridMARA methods, hence promoting reproducibility of results. Therefore, we recommend to clearly state adopted parameters if applying former methods for motion artifact removal.
Round 2
Reviewer 1 Report
I am happy that the authors addressed all my comments, questions and suggenstions and revised the draft accordingly - I would now approve the paper for publication. Kind regards.
Author Response
Dear Reviewer,
We would like to thank you once again for your precious inputs to our manuscript, making it much more complete and better to understand.